# Structure and function of the mycobacterial transcription initiation complex with the essential regulator RbpA

Elizabeth A Hubin[1], Allison Fay[2], Catherine Xu[1†], James M Bean[2], Ruth M Saecker[3], Michael S Glickman[2,4], Seth A Darst[1*], Elizabeth A Campbell[1*]

[1]The Rockefeller University, New York, United States; [2]Immunology Program, Sloan-Kettering Institute, New York, United States; [3]Independent Researcher, Madison, United States; [4]Division of Infectious Diseases, Memorial Sloan-Kettering Cancer Center, New York, United States

**Abstract** RbpA and CarD are essential transcription regulators in mycobacteria. Mechanistic analyses of promoter open complex (RPo) formation establish that RbpA and CarD cooperatively stimulate formation of an intermediate (RP2) leading to RPo; formation of RP2 is likely a bottleneck step at the majority of mycobacterial promoters. Once RPo forms, CarD also disfavors its isomerization back to RP2. We determined a 2.76 Å-resolution crystal structure of a mycobacterial transcription initiation complex (TIC) with RbpA as well as a CarD/RbpA/TIC model. Both CarD and RbpA bind near the upstream edge of the −10 element where they likely facilitate DNA bending and impede transcription bubble collapse. In vivo studies demonstrate the essential role of RbpA, show the effects of RbpA truncations on transcription and cell physiology, and indicate additional functions for RbpA not evident in vitro. This work provides a framework to understand the control of mycobacterial transcription by RbpA and CarD.

*For correspondence: darst@ rockefeller.edu (SAD); elizabeth. campbell0@gmail.com (EAC)

Present address: †Department of Chemistry, University of Cambridge, Cambridge, United Kingdom

Competing interests: The authors declare that no competing interests exist.

## Introduction

The bacterial pathogen *Mycobacterium tuberculosis* (*Mtb*) is the causative agent of tuberculosis, an ongoing world health problem. RNA polymerase (RNAP), responsible for all transcription in bacteria, is the target of the rifamycin class of antibiotics, a first-line therapeutic treatment for tuberculosis (*Chakraborty and Rhee, 2015*). Thus, RNAP is a proven drug target, highlighting the importance of gaining a structural and functional understanding of mycobacterial transcription - an understanding made difficult by the lack of a mycobacterial RNAP structure.

Bacterial transcription initiation is controlled by promoter-specificity σ-factors, which associate with the core RNAP ($\alpha_2\beta\beta'\omega$ subunits), generating the holoenzyme (holo; *Gruber and Gross, 2003*; *Murakami and Darst, 2003*). Promoter DNA sequences are recognized by holo (*Murakami et al., 2002*), triggering a series of conformational changes as the enzyme unwinds 12 to 14 base pairs of DNA to generate the transcription bubble and loads the template-strand (t-strand) DNA into the RNAP active site, resulting in the transcriptionally-competent open promoter complex (RPo; *Saecker et al., 2011*).

The vast majority of mechanistic studies on bacterial transcription initiation have used *Escherichia coli* (*Eco*) RNAP as a model. However, the properties of *Eco* RNAP are not necessarily representative of RNAPs from other bacterial species (*Davis et al., 2015*; *Schroeder and deHaseth, 2005*; *Whipple and Sonenshein, 1992*). In contrast to *Eco* holo, which forms an essentially irreversible RPo

on many promoters, the mycobacterial holo forms an unstable RPo with a half-life of a few minutes or less when compared with *Eco* holo on the same promoters (*Davis et al., 2015*). Two transcription factors, CarD and RbpA, both essential in mycobacteria but absent in *Eco*, potentiate the activity of the mycobacterial RNAP (*Forti et al., 2011*; *Stallings and Glickman, 2011*; *Tabib-Salazar et al., 2013*).

CarD associates with RNAP through its interaction with $\beta$-lobe 1 (also known as the protrusion; *Bae et al., 2015a*; *Stallings et al., 2009*) and stabilizes RPo by wedging a conserved Trp residue into the widened minor groove at the upstream edge of the transcription bubble (*Bae et al., 2015a*). RbpA forms a tight interaction with the $\sigma_2$ domain (−10 element recognition domain) of Group 1 and Group 2 $\sigma$ factors through its C-terminal $\sigma$-interacting domain (SID; *Bortoluzzi et al., 2013*; *Hubin et al., 2015*; *Tabib-Salazar et al., 2013*), but how other RbpA structural elements interact with the transcription initiation complex (TIC) and activate transcription is unknown.

ChIP-Seq studies indicate CarD is present at most, if not all, $\sigma^A$ promoters in mycobacteria (*Srivastava et al., 2013*). Comprehensive genomic data for RbpA occupancy in vivo is not available, but RbpA forms a very tight complex with $\sigma^A$ and $\sigma^A$-holo (*Hubin et al., 2015*) and is likely present at most $\sigma^A$ promoters as well. Thus, the structural mechanism for CarD and RbpA function must be compatible with the two transcription activators acting simultaneously on the mycobacterial TIC.

Here, we determined a 2.76 Å-resolution crystal structure of a mycobacterial TIC with RbpA. Structural analysis along with a combination of in vitro and in vivo functional approaches was used to extend our understanding of the roles of RbpA and CarD in regulating mycobacterial transcription. We show that RbpA and CarD cooperatively stimulate formation of an intermediate leading to RPo, a step defective in promoters lacking a −35 element, which represents the majority of mycobacterial promoters (*Cortes et al., 2013*). This work provides an unprecedented structural framework to understand mycobacterial transcription and insight into why RbpA is essential.

## Results

### Crystal structure of the *M. smegmatis* RbpA/TIC

Avirulent *M. smegmatis* (*Msm*) is not a viable model organism for *Mtb* pathogenesis but the *Msm* RNAP and associated transcription system is an excellent model for *Mtb* transcription (*Supplementary file 1*). To provide a structural basis for understanding mycobacterial transcription initiation, we crystallized and determined the 2.76 Å-resolution structure of a 446 kDa *Msm* TIC containing RbpA, $\sigma^A$-holo, and an upstream fork junction promoter fragment (*Figure 1*, *Figure 1—figure supplement 1A–C*, *Supplementary file 2*). Analysis of the structure provides several highlights that will be described elsewhere. Here we focus on the role of RbpA in the context of the TIC (*Figure 1*).

RbpA comprises four structural elements, the N-terminal tail (NTT), the core domain (CD), the basic linker (BL), and the SID (*Figure 1A*). The RbpA$^{SID}$ association with $\sigma^A_2$ matches closely the previously determined RbpA$^{SID}$/$\sigma^A_2$ crystal structure (PDB ID 4X8K; *Hubin et al., 2015*), with an rmsd of 0.451 Å over 152 C$\alpha$ atoms. N-terminal to the RbpA$^{SID}$, we built the RbpA$^{BL}$ and fit the previously determined structure of the *Mtb* RbpA$^{CD}$ (residues 26–69; PDB ID 2M4V; (*Bortoluzzi et al., 2013*); rmsd of 0.775 Å over 40 C$\alpha$ atoms after final refinement; *Figure 1—figure supplement 1C*) into clear electron density (*Figure 1—figure supplement 1B–C*), but density was absent for the 25-residue RbpA$^{NTT}$.

The RbpA$^{CD}$ makes extensive contacts with the $\beta'$ Zinc-binding domain ($\beta'$ZBD) as well as with residues flanking the $\beta'$zipper (*Figure 1*, *Figure 1—figure supplement 1C*), structural elements of the RNAP 'clamp' module (*Cramer et al., 2000*; *Gnatt et al., 2001*) near the N-terminus of $\beta'$ that are conserved among bacterial RNAPs (*Lane and Darst, 2010a*, *2010b*). The RbpA$^{CD}$/$\beta'$ interface buries a surface area of 615 Å$^2$. Together with the RbpA$^{SID}$/$\sigma^A_2$ interface (954 Å$^2$ buried surface area), RbpA forms a bipartite protein/protein interface with holo that buries a total surface area of 1569 Å$^2$. An alignment of RbpA from 890 bacterial genomes indicates that the CD is 55% identical, but residues that contact the ZBD and zipper are 80% identical (*Figure 1—figure supplement 1D*), indicating that the observed RbpA$^{CD}$/$\beta'$ interface is a conserved feature of the RbpA/RNAP interaction.

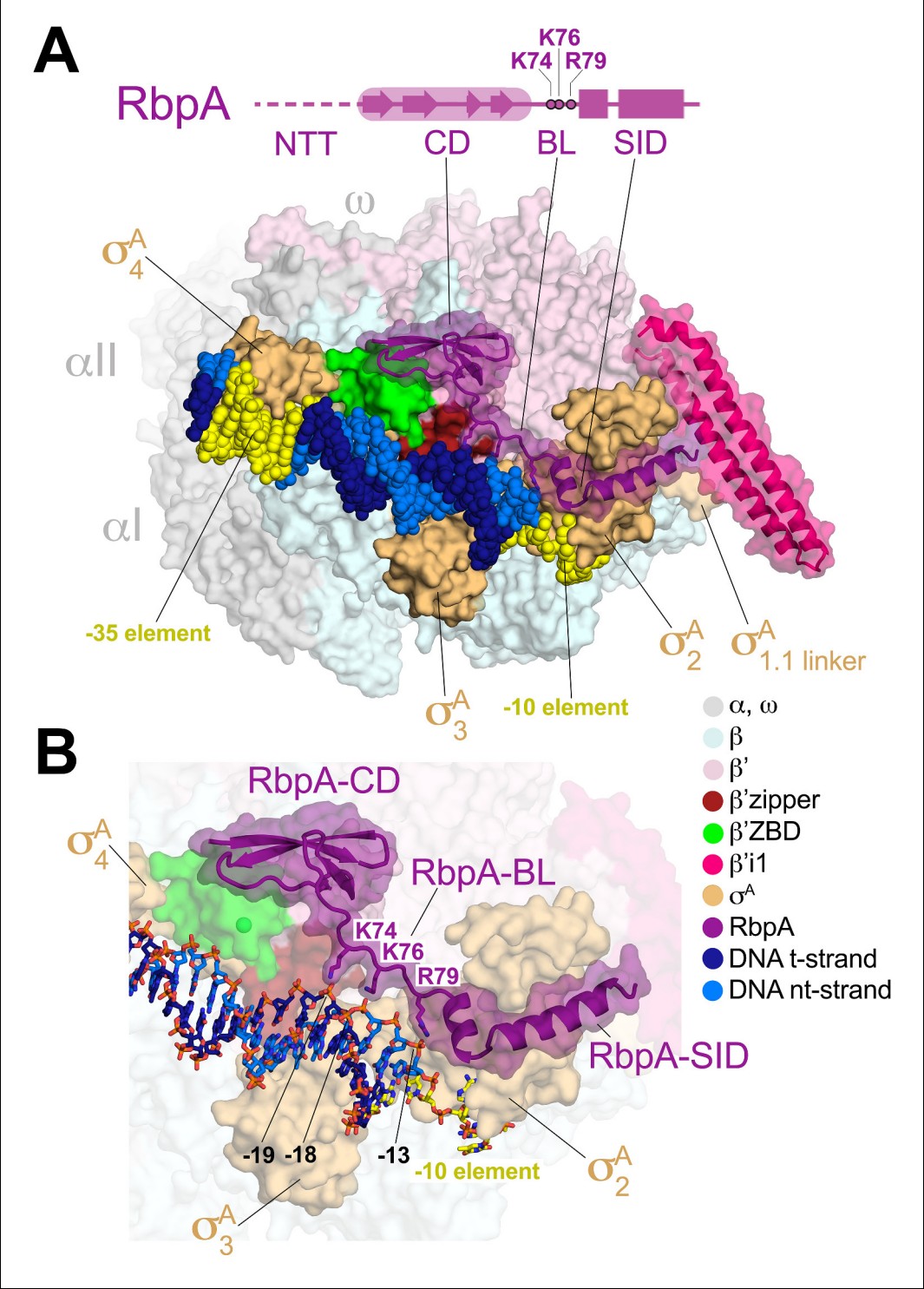

**Figure 1.** Structure of the *Msm* RbpA/TIC. (**A**) (top) The RbpA structural architecture is represented schematically. The CD is shown as a thick region, with β-strands represented as arrows. The α-helices of the SID are shown as rectangles. Linker regions lacking secondary structure, the NTT and BL, are represented by a thin line. The NTT is disordered in the crystal structure and is shown as a dashed line. Conserved basic residues in the BL (K74, K76, R79) that interact with the DNA phosphate backbone are denoted. (bottom) Overall structure of the *Msm* RbpA/TIC. The color-coding of most of the structural features is denoted in the legend. Protein components (core RNAP, σ^A, RbpA) are shown as molecular surfaces. The surfaces of RbpA and the lineage-specific insert β'i1 are transparent, revealing the α-carbon backbone ribbon underneath. RbpA side chains K74, K76, and R79 are shown

*Figure 1 continued on next page*

*Figure 1 continued*

in stick format. The DNA is shown as CPK atoms, with the −35 and −10 elements colored yellow. (**B**) Magnified view of the region including RbpA and the promoter DNA near the −10 element. The DNA is shown in stick format. The $\beta'$ZBD surface is transparent with the $Zn^{2+}$-ion shown as a sphere.

The following figure supplement is available for figure 1:

**Figure supplement 1.** Crystallization oligonucleotides, electron density maps, and RbpA sequence conservation.

---

Although the $RbpA^{CD}/\beta'$ interactions may modulate transcription initiation, such a role is not evident solely from examination of the structure. What is evident from the structure is that the RbpA/ RNAP interactions position the $RbpA^{BL}$ (residues 70–79) to interact with the DNA phosphate backbone just upstream of the −10 element (*Figure 1B*).

## RbpA-R79 anchors the DNA upstream of the transcription bubble

In the *Msm* RbpA/TIC, absolutely conserved RbpA-R79 (*Figure 1—figure supplement 1D*) makes a close (2.2 Å) polar interaction with the negatively charged phosphate between the −13 and −14 positions of the non-template strand (nt-strand) DNA (*Figure 1B*, *Figure 1—figure supplement 1B*), explaining previous findings that RbpA-R79 was important for the ability of RbpA to increase the overall affinity of mycobacterial holo to promoter DNA and for normal RbpA function in vivo (*Hubin et al., 2015*). The 10-residue mycobacterial $RbpA^{BL}$ harbors four positively charged residues (K73, K74, K76, R79; *Figure 1—figure supplement 1D*). In addition to the R79/-13 nt-strand DNA phosphate interaction, RbpA-K74 and K76 are also positioned to make long-range electrostatic interactions (5.4 and 5.5 Å, respectively) with the t-strand DNA phosphates between the −18/−19 and −17/−18 positions, respectively (*Figure 1B*, *Figure 1—figure supplement 1B*). Like RbpA-R79, RbpA-K76 is absolutely conserved among RbpA orthologs (*Figure 1—figure supplement 1D*).

## The RbpA-R79/DNA interaction is critical for RbpA in vitro function

To better understand the roles of the RbpA structural elements in transcription activation, we generated a series of RbpA N-terminal truncations, sequentially deleting the NTT ($RbpA^{CD-BL-SID}$) and the CD ($RbpA^{BL-SID}$; *Figure 2A*) and tested their function in abortive initiation assays along with $RbpA^{R79A}$ (*Figure 2B*, *Figure 2—figure supplement 1*). To expand upon previous *in vitro* studies (*Davis et al., 2015*; *Hubin et al., 2015*; *Rammohan et al., 2016, 2015*; *Srivastava et al., 2013*), we used $\sigma^A$, RbpA and CarD from *Mtb/M. bovis* (the *Mtb* and *M. bovis* factors are identical in sequence; *Supplementary file 1*) with recombinant *M. bovis* (*Mbo*) RNAP (99.9% identical in sequence to *Mtb* RNAP; *Supplementary file 1*; *Czyz et al., 2014*). We examined two previously characterized *Mtb* promoters: the *vapB10p* antitoxin promoter (VapB; *Cortes et al., 2013*) and the *rrnAP3* promoter (AP3; *Figure 2—figure supplement 1A*; *Gonzalez-y-Merchand et al., 1996*). ChIP-Seq studies in *Msm* indicated that CarD was present at essentially all promoters throughout the genome (*Srivastava et al., 2013*), suggesting that CarD and RbpA likely function together. We therefore performed these assays with and without CarD.

On VapB without CarD, RbpA and $RbpA^{CD-BL-SID}$ activated transcription ~2 fold (*Figure 2B*, columns 2–3), suggesting the $RbpA^{NTT}$ does not play a role in activation on this promoter. $RbpA^{BL-SID}$ activated ~3 fold (column 4), while $RbpA^{R79A}$ showed no activation (column 5), suggesting that the BL interaction with DNA is primarily responsible for activation (*Figure 1B*). CarD alone increased transcription 3-fold over holo (column 6), but the addition of RbpA increased activation to more than 6-fold (column 7), suggesting the activators work together. The RbpA mutants displayed similar effects without CarD, with the exception that $RbpA^{R79A}$ with CarD repressed transcription compared to CarD alone (column 10).

On AP3 without CarD, RbpA and derivatives behaved similarly as on VapB, but the repressive effect of $RbpA^{R79A}$ was noticeable in the absence of CarD (*Figure 2B*, column 14). With CarD, RbpA had little or no additional effect (columns 15–19). Like most promoters in *Mtb*, VapB lacks a −35 element (*Cortes et al., 2013*), while AP3 harbors a nearly consensus −35 element (*Figure 2—figure supplement 1A*). We hypothesized that RbpA may show increased potency on promoters lacking a

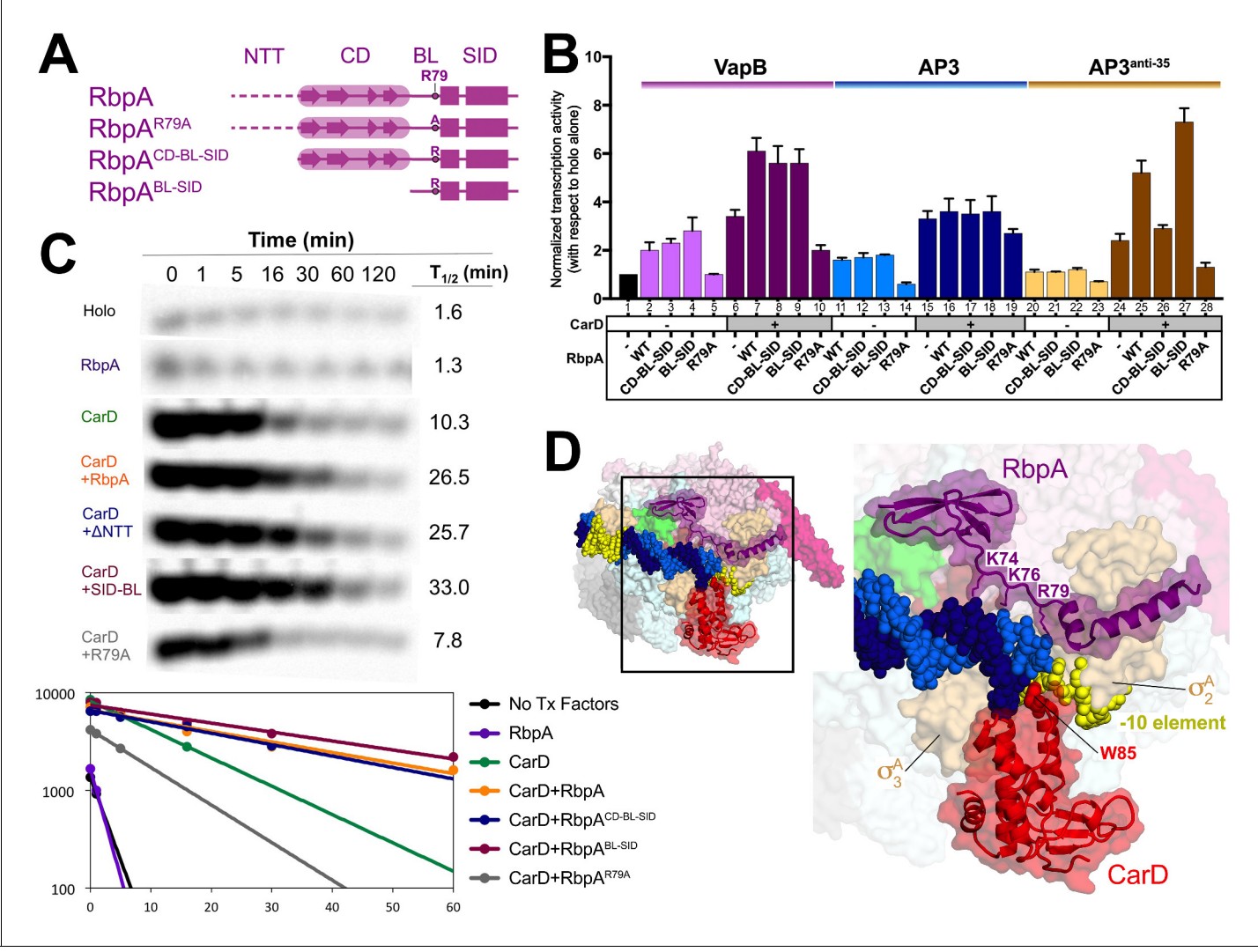

**Figure 2.** Function of RbpA and RbpA derivatives in transcription initiation and cooperativity with CarD. (A) Schematic diagram denoting the RbpA derivatives used in subsequent assays. (B) The effect of RbpA or RbpA derivatives (denoted at the bottom) on activation of abortive initiation from three different promoters (VapB, AP3, AP3$^{anti-35}$, denoted at the top) with or without CarD. The transcription activity for each promoter was normalized with respect to holo activity on that promoter (holo alone on each promoter was normalized to a value of 1, as shown in lane 1). The error bars denote the standard error from a minimum of three experiments. (C) Promoter complex lifetimes measured by abortive initiation on the AP3 promoter. In the top panel, [$^{32}$P]-labeled abortive transcript production at times after addition of a large excess of competitor promoter trap DNA (*Davis et al., 2015*) was monitored by polyacrylamide gel electrophoresis and autoradiography. On the bottom, transcript production was quantified by phosphorimagery and plotted. The lines indicate single-exponential decay curves fit to the data points. The calculated decay half-lives (t$_{1/2}$) are shown to the right of the gel images. (D) Structural model showing the *Msm* RbpA/TIC (color coded as in *Figure 1*) along with CarD (red), superimposed by aligning the thermus CarD/RPo structure (PDB ID 4XLR; (*Bae et al., 2015a*).

The following figure supplement is available for figure 2:

**Figure supplement 1.** Transcription templates and abortive initiation data.

−35 element. To test this, we generated an AP3 derivative, AP3$^{anti-35}$, containing the least likely base to occur at each position of the −35 element (*Figure 2—figure supplement 1A*; *Shultzaberger et al., 2007*). Overall, transcription activity on AP3$^{anti-35}$ was decreased more than 10-fold compared to AP3 (*Figure 2—figure supplement 1C*), and transcription without CarD was barely measurable. However, RbpA in combination with CarD showed a cooperative effect, with a

greater than multiplicative effect on transcription activation (*Figure 2B*, compare columns 20 and 24 to 25), suggesting the absence of a −35 element accentuates the RbpA/CarD cooperative effect.

In summary, we found that in vitro: (1) Deletion of the RbpA$^{NTT-CD}$ was not deleterious to RbpA activation function, and (2) RbpA-R79 was required for activation function. These results suggest the RbpA$^{BL-SID}$ provides the critical contacts for transcription activation and imply that the role of the SID is to localize the BL to contact the DNA.

Mycobacterial RNAP forms unstable RPo compared to *Eco* when compared on the same promoters, and the main function of CarD appears to be to stabilize RPo (*Bae et al., 2015a*; *Davis et al., 2015*; *Rammohan et al., 2015*). We examined the role of RbpA in stabilizing RPo (with and without CarD) with promoter lifetime assays on AP3 (*Davis et al., 2015*). RbpA on its own had little to no impact on the RPo half-life (t$_{1/2}$), but the combination of RbpA with CarD had a strongly cooperative effect, extending t$_{1/2}$ more than 2-fold compared to CarD alone (*Figure 2C*). With CarD, RbpA$^{CD-BL-SID}$ had a similar effect as RbpA, while RbpA$^{BL-SID}$ extended t$_{1/2}$ to an even greater extent than RbpA. On the other hand, RbpA$^{R79A}$ shortened t$_{1/2}$ significantly below CarD alone, consistent with our findings suggesting that the RbpA$^{CD}$ has a negative effect on transcription.

Our results (*Figure 2*) indicate that RbpA and CarD function cooperatively, in agreement with recent studies of *Rammohan et al. (2016)*. To evaluate whether occupancy of the TIC by RbpA and CarD simultaneously was structurally feasible, we placed CarD in the context of the *Msm* RbpA/TIC by superimposing corresponding Cα coordinates of the thermus CarD/RPo structure (PDB ID 4XLR; *Bae et al., 2015a*). While CarD and RbpA interact with the promoter DNA at or near the upstream part of the −10 element, they do so from opposite sides of the DNA (*Figure 2D*) without any steric clash.

## RPo formation by mycobacterial RNAP requires three steps

Formation of RPo is a multi-step process (*Saecker et al., 2011*). A sequential mechanism involving at least three steps has been proposed for *Eco* σ$^{70}$-holo based on several decades of studies using the lacUV5 (*Buc and McClure, 1985*), λP$_R$ (*Roe et al., 1985*; *Saecker et al., 2011*), T7A1 (*Sclavi et al., 2005*), and *rrnB* P1 (*Rutherford et al., 2009*) promoters:

$$\mathrm{R+P} \underset{k_{-1}}{\overset{k_1}{\rightleftharpoons}} \mathrm{RP1} \underset{k_{-2}}{\overset{k_2}{\rightleftharpoons}} \mathrm{RP2} \underset{k_{-3}}{\overset{k_3}{\rightleftharpoons}} \mathrm{RPo} \qquad (1)$$

In *Equation 1*, R and P represent free RNAP holo and promoter DNA, respectively, and RPo is the final, transcription-competent open complex. RP1 and RP2 represent kinetically-significant intermediates along the pathway of RPo formation. It is thought that RPo formation at most promoters involves similar intermediates, but the significance of different intermediates in each case is dictated by the promoter sequence (*Saecker et al., 2011*). This pathway (*Equation 1*) has been studied using mostly approaches that require the RPo reactions to be halted, then probed after the fact (*Ross and Gourse, 2009*; *Saecker et al., 2003*).

Does *Equation 1* describe RPo formation by mycobacterial RNAP? We monitored the kinetic steps of RPo formation by mycobacterial RNAP, and the effects of RbpA and CarD on those steps, using a real-time fluorescence assay first reported by *Ko and Heyduk (2014)*. In this relatively non-perturbing assay, the fluorescence of a Cy3 fluorophore attached to the promoter DNA at the nt-strand +2 position is monitored as RPo formation takes place (*Ko and Heyduk, 2014*; *Rammohan et al., 2015, 2016*) (*Figure 3—figure supplement 1A*). Detecting forward progress in this assay does not depend on the use of competitors, such as heparin. Because interactions between the +2 position of the promoter and RNAP undergo dramatic changes as RPo forms, the environmentally sensitive Cy3 fluorophore (*Aramendia et al., 1994*; *Toutchkine et al., 2003*) is well suited to report on the formation of intermediates in the pathway (*Ko and Heyduk, 2014*; *Rammohan et al., 2015, 2016*).

Changes in fluorescence as a function of time were monitored after rapid mixing of the Cy3-AP3 promoter construct (Cy3-AP3; *Figure 3—figure supplement 1A*) with a given protein sample (*Eco* holo, *Mbo* holo, *Mbo* holo+RbpA, *Mbo* holo+CarD, or *Mbo* holo+RbpA+CarD; *Figure 3A–C*, *Figure 3—figure supplement 1B–C*). To separate the binding step (dependent on [RNAP]) from subsequent conversions ([RNAP]-independent, see *Equation 1*), mixing was done for a series of [RNAP]

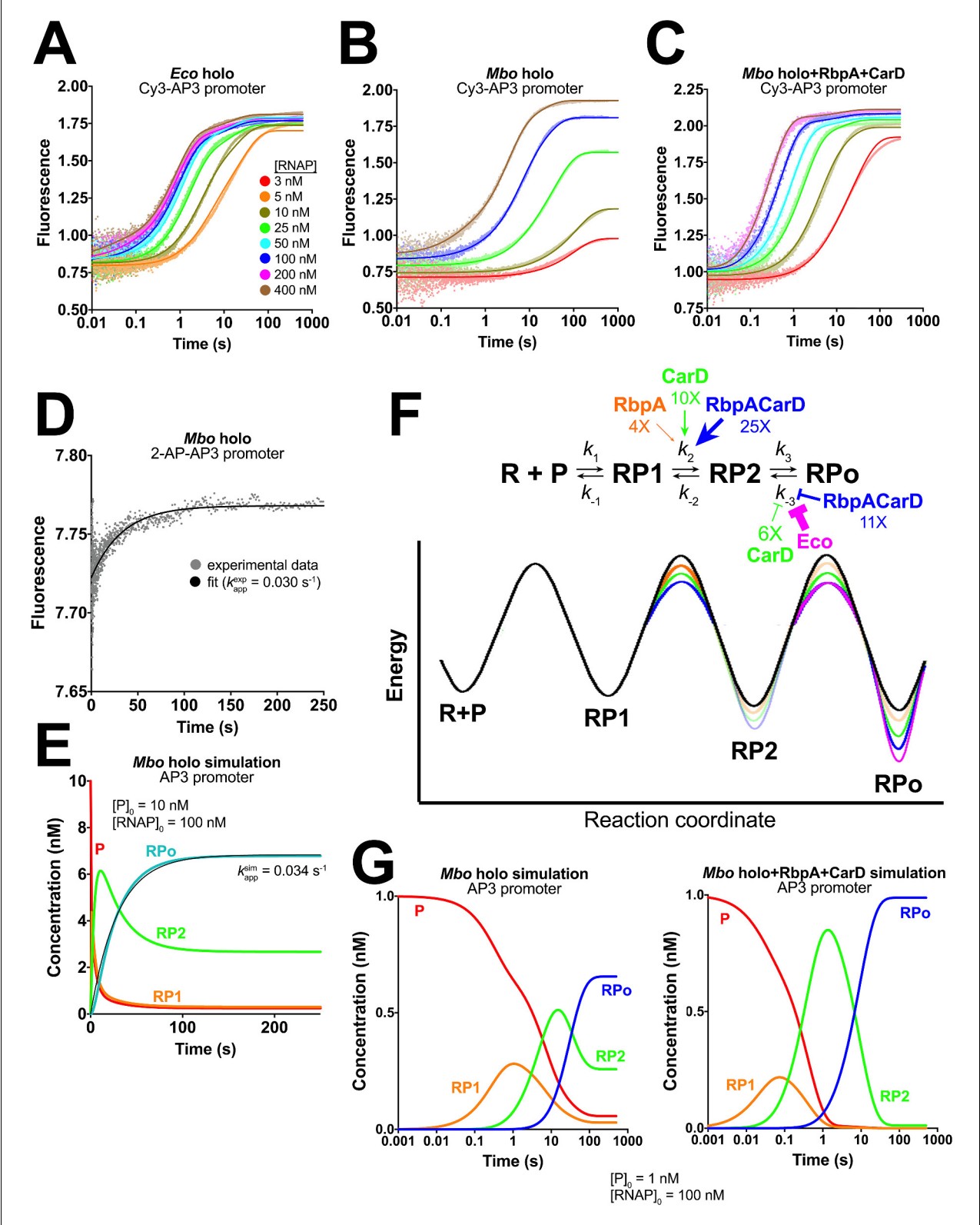

**Figure 3.** Kinetics of RPo formation on the AP3 promoter. (**A**) Plot showing the fluorescence signal vs. time after rapid mixing of *Eco* holo with Cy3-AP3 promoter (*Figure 3—figure supplement 1A*) in a stopped flow fluorimeter. The [RNAP] giving rise to each curve is color-coded as shown in the legend. The experimental data are shown as points. The data were fit using the three-step sequential kinetic scheme (*Equation 1*), yielding the parameters listed in *Supplementary file 4*. The curve fits are shown as solid lines. (**B**) Same as (**A**) but with *Mbo* holo. (**C**) Same as (**A**) but with *Mbo*

*Figure 3 continued on next page*

*Figure 3 continued*

holo+RbpA+CarD. (**D**) Plot showing the fluorescence signal vs. time after mixing 100 nM *Mbo* holo with 10 nM 2-AP-AP3 promoter (*Figure 3—figure supplement 1G*) in a stopped flow fluorimeter. The experimental data are shown as points. The data were fit to a single-exponential (solid black line): $F = F_0 + (F_{max} - F_0)(1 - e^{k_{app}^{exp} t})$ yielding $k_{app}^{exp} = 0.030$ s$^{-1}$. (**E**) Simulation of changes in the populations of P (red), RP1 (orange), RP2 (green), and RPo (blue) under the same conditions as the experiment of panel (**D**). The kinetic parameters used to generate the simulation are from *Supplementary file 4*. The data were fit to a single-exponential (thin black line), yielding $k_{app}^{sim} = 0.034$ s$^{-1}$. (**F**) (top) The three-step sequential kinetic scheme that best accounts for all of the kinetic data is shown. The steps targeted by the transcription factors RbpA (orange), CarD (green), or RbpACarD together (blue) are denoted. Arrows pointing at a parameter indicate an increase in that parameter in the presence of the factor (compared to *Mbo* holo alone by the fold-amount shown below); the 'T' symbol indicates the factor reduces the parameter. The most important difference between the reference (*Mbo* holo alone) and *Eco* holo (magenta) is also illustrated. RbpA, CarD, and RbpACarD all increase $k_2$ significantly. CarD also reduces $k_{-3}$, as does *Eco* holo to a much greater extent. (bottom) Schematic free energy profile for RPo formation. The black curve represents *Mbo* holo alone. The colored curves illustrate the most important changes induced by the factors (RbpA, orange; CarD, green; RbpACarD, blue; *Eco* holo, magenta). (**G**) Simulations of changes in the populations of P (red), RP1 (orange), RP2 (green), and RPo (blue) when $[P]_0 = 1$ nM and $[RNAP]_0 = 100$ nM for *Mbo* holo alone (left) and *Mbo* holo+RbpA+CarD (right). The kinetic parameters used to generate the simulation are listed in *Supplementary file 4*. RbpA and CarD together induce a significant increase in $k_2$, producing a large transient burst of RP2, driving formation of RPo.

The following figure supplement is available for figure 3:

**Figure supplement 1.** Cy3 promoters, association and dissociation data, activation energies, and 2-AP promoter.

(over ~100 fold range; *Supplementary file 3*). When factors (RbpA or CarD) were present, they were well above saturating concentrations to ensure high occupancy of RNAP and promoter-bound species (*Supplementary file 3*).

## Qualitative analysis of kinetic data for the AP3 promoter

Some important conclusions can be drawn from examination of the raw fluorescence traces (*Figure 3A–C*, *Figure 3—figure supplement 1B–C*) even without quantitative analysis:

1. At a given [RNAP], *Eco* holo reaches a plateau value about 5-fold faster than *Mbo* holo (*Figure 3A–B*).
2. For *Eco* holo, the fluorescence plateau value is independent of [RNAP]; even at the lowest [RNAP], *Eco* holo eventually reaches the maximum plateau value (*Figure 3A*), On the other hand, the plateau fluorescence for *Mbo* holo depends strongly on [RNAP]; lower [RNAP] plateaus at a lower fluorescence value (*Figure 3B*). These data indicate that *Eco* RNAP converts all of the Cy3-AP3 to RPo in an essentially irreversible reaction, whereas *Mbo* RPo manifests profound reversibility (*Davis et al., 2015*); *Mbo* RPo coexists with R, P, and intermediates at equilibrium (*Equation 1*).
3. Adding RbpA (*Figure 3—figure supplement 1B*), CarD (*Figure 3—figure supplement 1C*), and finally both factors together (*Figure 3C*) qualitatively moves the behavior of the *Mbo* system to that of *Eco* holo.

## Quantitative analysis of kinetic data for the AP3 promoter

We analyzed the fluorescence progress curves (*Figure 3A–C*, *Figure 3—figure supplement 1B–C*) by globally fitting the data for each concentration series collected on a given day to a kinetic mechanism (*Johnson et al., 2009b*, *2009a*). We assumed that the different RNAPs (*Eco* or *Mbo*) or presence of the factors determined the rate constants associated with the mechanism, but did not alter the basic mechanism. Based on previous analyses (*Buc and McClure, 1985*; *Roe et al., 1985*; *Rutherford et al., 2009*; *Saecker et al., 2011*; *Sclavi et al., 2005*), we considered sequential (linear) kinetic schemes (Appendix). The three-step sequential model best accounted for all of the kinetic data (*Supplementary file 4*, Appendix), consistent with previous work with *Eco* RNAP.

We can make several observations regarding the validity of our approach and some proposals based on the resulting estimates of the kinetic parameters (*Supplementary file 4*):

1. The promoter half-lives calculated using the fitted kinetic parameters (*Supplementary file 4*; *Tsodikov and Record, 1999*) match the experimentally determined values (*Figure 2C*, *Supplementary file 4*), providing very strong support for the kinetic analysis.

2. Structural analysis indicates that RbpA and CarD would be unlikely to influence the Cy3 fluorescence in any of the RNAP/DNA-bound states, and the fluorescence scale factors describing the contribution of each fluorescent species to the total signal (a, b, c, d) refine to similar values for all of the samples (*Supplementary file 4*). Thus, these values may be physically meaningful. The scale factor for P ('a') refines to the lowest value (~0.27 ± 0.02) where the fluorophore would be the most solvent exposed and would be expected to have the lowest fluorescence intensity. The scale factor for RP1 ('b') refines to an intermediate value (~0.45 ± 0.06), suggesting the fluorophore environment is more proteinacous but is still relatively solvent exposed. The scale factors for RP2 ('c') and RPo ('d') refine to the highest values (~1.2 ± 0.04, ~1.2 ± 0.02, respectively) suggesting that in these states, the fluorophore is relatively shielded from solvent (i.e. inside the RNAP active site cleft).

3. In addition to a, b, c, d, the value of $k_3$ also refined to similar values across all of the samples (the standard deviation of the average value across all of the samples is only 26%; *Supplementary file 4*). This rate may not be strongly influenced by the identity of the holo (*Eco* or *Mbo*) nor by the presence of RbpA and/or CarD, suggesting that $k_3$ may be primarily influenced by the properties of the DNA.

4. For the two most active samples (*Eco* holo and *Mbo* holo+RbpA+CarD), $k_1$ refines to a maximum value of $1.2 \times 10^8$ $M^{-1}s^{-1}$, suggesting the possibility that the forward rate for these bimolecular reactions may be diffusion limited. If this is the case, it suggests that the formation of RP1 does not require large conformational changes in the RNAP or the DNA.

For *Mbo* holo and *Mbo* holo+RbpA, the short RPo half-lives (*Figure 2C*) made it feasible to monitor RPo dissociation using the fluorescence assay (*Figure 3—figure supplement 1D–E*). The observed dissociation rates were consistent with the RPo $t_{1/2}$ measured by abortive initiation (*Figure 2C*) and calculated from the fitted kinetic constants (*Supplementary file 4*).

Although these observations all point to the veracity of our kinetic analysis, some of the fitted parameters were not well constrained (*Johnson et al., 2009b*). We imposed additional constraints on the data to reduce the degrees of freedom for the fitted parameters. The details of the fitting and validation procedures are described in the Appendix. With these additional constraints, the fits for the parameters were well constrained with the exception of $k_1$ for *Mbo* holo+RbpA+CarD (Appendix). The final kinetic parameters (*Table 1*) were close to the original, unconstrained values (compare *Table 1* with *Supplementary file 4*).

## RbpA and CarD cooperate to affect two distinct steps of RPo formation on the AP3 promoter

The schematic energy profiles for RPo formation (*Figure 3F*) help illustrate the following conclusions from our analysis on AP3:

1. The addition of RbpA alone to *Mbo* holo has a moderate effect on the kinetic parameters, lowering the energy barrier for the RP1 -> RP2 transition (increasing $k_2$ ~4 fold), consistent with the small effect of RbpA on transcription activity (*Figure 2B*).

2. The addition of CarD alone also lowers the energy barrier for the RP1 -> RP2 transition (increasing $k_2$ ~10 fold).

3. The effect of RbpA and CarD together on increasing both the rate of formation and the final amount of *Mbo* RPo (compared to *Mbo* holo alone) can be understood by considering fluxes into and out of the intermediate RP2. RbpA and CarD together have a cooperative effect on the RP1 -> RP2 transition, increasing $k_2$ ~25 fold, but have little effect on $k_{-2}$, increasing $K_2$ more than 20-fold (*Table 1*). The factors have little influence on $k_3$, so the net effect is a burst of RP2 in the presence of RbpA and CarD that drives the formation of RPo by mass action (*Figure 3G*).

4. Once formed, RPo is stabilized by CarD (decreasing $k_{-3}$ ~6 fold). RbpA has no detectable effect on $k_{-3}$ at AP3. However, when RbpA and CarD are present together, $k_{-3}$ is reduced ~11 fold.

5. The main difference between *Eco* and *Mbo* holos is the very large energy barrier for the conversion of RPo -> RP2 for *Eco* holo (*Figure 3—figure supplement 1F*). This accounts for the essentially irreversible RPo formed by *Eco* holo compared to the relatively unstable RPo of *Mbo* holo observed here and in other studies (*Davis et al., 2015*).

6. The rate of formation of RP2 ($k_2$) for *Mbo* holo+RbpA+CarD is nearly an order of magnitude higher than that of *Eco* holo (9.2 $s^{-1}$ vs 1.2 $s^{-1}$, respectively, *Table 1*), indicating that the strong activity of *Eco* holo is attributable almost totally to its very stable RPo (very small $k_{-3}$).

**Table 1.** Constrained kinetic parameters on the *Mtb* AP3 promoter. The fluorescence progress curves (*Figure 3A–C*, *Figure 3—figure supplement 1B–C*) were fit according to the 3-step sequential kinetic scheme:

$$\mathrm{R+P} \underset{k_{-1}}{\overset{k_1}{\rightleftarrows}} \mathrm{RP1} \underset{k_{-2}}{\overset{k_2}{\rightleftarrows}} \mathrm{RP2} \underset{k_{-3}}{\overset{k_3}{\rightleftarrows}} \mathrm{RPo}$$

| parameter[***] | RNAP *Mbo* holo | +RbpA | + CarD | +CarD+RbpA | *Eco* holo |
|---|---|---|---|---|---|
| n[*] | 5[†] | 4[‡] | 2 | 2 | 1 |
| $k_1$ ($M^{-1}s^{-1}$) | $1.1 \times 10^7$ | $1.7 \times 10^7$ | $4.4 \times 10^7$ | $1.2 \times 10^8$ | $1.2 \times 10^8$ |
| $k_{-1}$ ($s^{-1}$) | 2.1 | 1.3 | 3.6 | 23 | 3.4 |
| $K_1$ ($M^{-1}$)[§] | $5.2 \times 10^6$ | $1.3 \times 10^7$ | $1.2 \times 10^7$ | $>5.2 \times 10^6$ | $>3.5 \times 10^7$ |
| $k_2$ ($s^{-1}$) | 0.36 | 1.3 | 3.5 | 9.2 | 1.2 |
| $k_{-2}$ ($s^{-1}$) | 0.041 | 0.13 | 0.076 | 0.046 | 0.11 |
| $K_2$ | 8.8 | 10 | 46 | 200 | 11 |
| $k_3$ ($s^{-1}$) | 0.035 | 0.082 | 0.066 | 0.11 | 0.083 |
| $k_{-3}$ ($s^{-1}$) | 0.014 | 0.013 | **$2.3 \times 10^{-3}$** | **$1.3 \times 10^{-3}$** | **0** |
| $K_3$ | 2.5 | 6.3 | 29 | 85 | - |
| $K_1K_2K_3$ ($M^{-1}$) | $1.2 \times 10^8$ | $8.3 \times 10^8$ | $1.6 \times 10^{10}$ | $8.8 \times 10^{10}$ | - |
| $k_d$ ($s^{-1}$)[¶] | $6.3 \times 10^{-3}$ | $7.1 \times 10^{-3}$ | $1.2 \times 10^{-3}$ | $3.7 \times 10^{-4}$ | - |
| $t_{1/2}$ (min)[¶] | 1.8 | 1.6 | 9.9 | 31 | - |
| $t_{1/2}^{exp}$ (min)[**] | ~2 | ~1.5 | ~10 | ~30 | >>60 |

Color coding:

Grey: 5–10-fold > *Mbo* holo; Pink: 5–10-fold < *Mbo* holo

Green: > 10 fold over *Mbo* holo; Red: more than 10-fold < *Mbo* holo

Bold text denotes that that parameter was fixed during the refinement (see Appendix).

[***]Because the independent trials for each sample were analyzed together, we could not calculate errors in the fitted parameters across trials. The standard errors from the fits are likely to be underestimates of the errors (Johnson et al., 2009a; 2009b). Therefore, Table 1 does not report errors in the fitted parameters and we presume the errors are around 10–15%, as seen in the unconstrained analysis (Supplementary file 4).

[*]Number of independent trials.

[†]Includes three association series (*Figure 3B*), one dissociation experiment (*Figure 3—figure supplement 1D*), and the 2-AP experiment (*Figure 3E*).

[‡]Includes three association series (*Figure 3—figure supplement 1B*) and one dissociation experiment (*Figure 3—figure supplement 1E*).

[§]The values for $K_1$, $K_2$, and $K_3$ were calculated from the fitted parameters: $K_1 = k_1/k_{-1}$, $K_2 = k_2/k_{-2}$, $K_3 = k_3/k_{-3}$.

[¶]The value for $k_d$, the dissociation rate for RPo, was calculated using equation (17) of *Tsodikov and Record (1999)*:

$$\frac{1}{k_d} = \frac{1}{k_{-3}} + \frac{1 + K_3}{k_{-2}} + \frac{K_2 + K_2K_3}{k_{-1}} + \frac{1}{k_{-1}} \tag{2}$$

The value for $t_{1/2}$ was calculated as $t_{1/2} = \ln(2)/k_d$.

[**]The experimental half-life ($t_{1/2}^{exp}$) was determined from promoter lifetime experiments (*Figure 2C*).

## DNA opening (RP2 -> RPo) and closing (RPo -> RP2) steps are rate-limiting at the AP3 promoter

Analysis of the activation energies required to traverse the transition states for each kinetic step indicates that for all of the samples on the AP3 promoter, the rate-limiting step (the highest activation energy) in the forward direction is the conversion of RP2 -> RPo, while the same step is rate-limiting in the reverse direction (RPo -> RP2; *Figure 3—figure supplement 1F*). Consistent with previous analyses, we hypothesized that this rate-limiting step in the forward direction corresponds to the formation of the full transcription bubble (*Saecker et al., 2011*). We tested this hypothesis by monitoring the time-dependent fluorescence from an AP3 promoter derivative harboring 2-aminopurine (2-AP) at the t-strand +2 position (*Figure 3—figure supplement 1G*) during RPo formation. 2-AP forms a Watson-Crick base pair with thymine in the context of normal B-form DNA (*Nordlund et al., 1989*). 2-AP fluorescence is strongly quenched by stacking interactions with neighboring bases, such as in the context of duplex DNA (*Jean and Hall, 2001*; *Ward et al., 1969*), making 2-AP an excellent probe for transcription bubble formation (*Lim et al., 2001*; *Roy, 2003*). Available data suggest that transcription bubble formation initiates within the −10 element then propagates downstream (*Chen and Helmann, 1997*; *Heyduk et al., 2006*; *Lim et al., 2001*). We placed 2-AP at the very downstream edge of the transcription bubble so that an increase of 2-AP fluorescence signals full transcription bubble formation, directly preceding or concurrent with RPo formation.

An increase of 2-AP AP3 fluorescence was observed upon the addition of *Mbo* holo (*Figure 3D*). We fit the data to a single-exponential to obtain an apparent rate constant, $k_{app}^{exp}$ = 0.030 s$^{-1}$ (*Figure 3D*), nearly identical to the rate of RPo formation simulated from the independently determined kinetic parameters (*Table 1*; *Figure 3E*; $k_{app}^{sim}$ = 0.034 s$^{-1}$), suggesting that the rate-limiting RP2 -> RPo transition involves the formation of the full transcription bubble.

## The −35-less VapB promoter has a very slow RP1 -> RP2 transition that is rescued by RbpA

It has been noted that RbpA has a more potent effect on transcription of the *Mtb* VapB promoter than on AP3 (*Hubin et al., 2015*). In our assays this is most notable with CarD (*Figure 2B*). In order to probe the effects of RbpA and RbpA truncations on transcription in more detail, we generated the Cy3-VapB promoter fragment (*Figure 3—figure supplement 1A*) and monitored RPo formation with the fluorescence assay (*Supplementary file 5*). Although both AP3 and VapB are *bona fide Mtb* promoters, VapB is more typical in that it lacks recognizable elements upstream of the −10 element (*Cortes et al., 2013*).

*Mbo* holo forms RPo on VapB about 10 times more slowly than on AP3 (*Figure 4A*). The VapB kinetic data were also best fit by the sequential three-step model (*Equation 1*). When comparing *Mbo* holo, *Mbo* holo+RbpA, and *Eco* holo on the two promoters, the consistently significant difference in the kinetic parameters is that the RP1 -> RP2 transition ($k_2$) is much slower on VapB; the VapB $k_2$ is 7-fold (*Mbo* holo+RbpA), 47-fold (*Mbo* holo alone), or 230-fold (*Eco* holo) smaller than the AP3 $k_2$ (*Supplementary file 6*; *Figure 4B*). Like on AP3 (*Table 1*; *Figure 3F*), RbpA targets the RP1 -> RP2 transition, increasing $k_2$ on VapB by more than 20-fold (*Supplementary file 6*; *Figure 4C*).

## Full-length RbpA is essential for normal growth of *Msm*

The increased in vitro activity of RbpA$^{SID-BL}$ (*Figure 2B–C*) led us to investigate whether the NTT and CD were required for the essential functions of RbpA in vivo. We constructed an *Msm* strain in which the *rbpA* gene could be efficiently swapped by allelic exchange. As *rbpA* was previously shown to be essential in mycobacteria (*Bortoluzzi et al., 2013*; *Forti et al., 2011*; *Sassetti et al., 2003*), we generated a *ΔrbpA* allele in a strain carrying a second copy of *rbpA* integrated at the *attB* chromosomal site conferring streptomycin resistance (MGM6228), which can be efficiently swapped for *rbpA* alleles on an *attB* integrating plasmid conferring kanamycin resistance. In accordance with the essential role of *rbpA*, swapping a vector control plasmid for *rbpA* yielded no transformants, in contrast to efficient marker exchange seen with full length RbpA. We next tested *rbpA* mutant alleles encoding the RbpA derivatives tested in vitro (*Figure 2A*). RbpA$^{CD-BL-SID}$, RbpA$^{BL-SID}$, and RbpA$^{R79A}$ all supported viability. Although similar frequencies of allelic replacement were observed for all three alleles, both RbpA$^{BL-SID}$ and RbpA$^{R79A}$ transformants were smaller than those of RbpA$^{CD-BL-SID}$. To

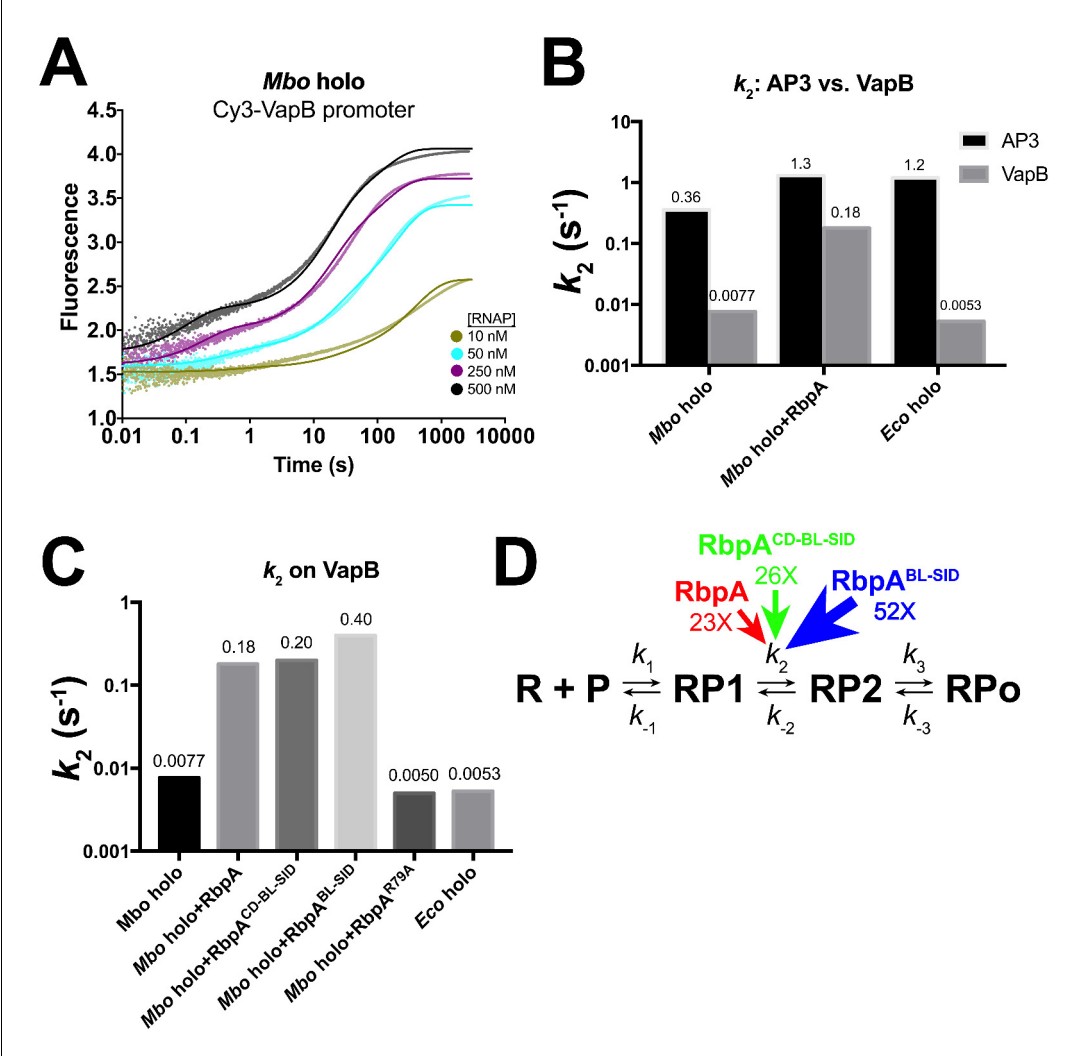

**Figure 4.** Kinetics of RPo formation on the VapB promoter. (**A**) Plot showing the fluorescence signal vs. time after rapid mixing of *Mbo* holo with Cy3-VapB promoter (*Figure 3—figure supplement 1A*) in a stopped flow fluorimeter. The [RNAP] giving rise to each curve is color-coded as shown in the legend. The experimental data are shown as points. The data were fit using the three-step sequential kinetic scheme (*Equation 1*), yielding the parameters listed in *Supplementary file 8*. The curve fits are shown as solid lines. (**B**) Bar graph comparing the values of $k_2$ for *Mbo* holo, *Mbo* holo +RbpA, and *Eco* holo on the AP3 promoter (black bars) and the VapB promoter (grey bars). (**C**) Bar graph comparing the values of $k_2$ for the denoted samples on the VapB promoter. (**D**) The three-step sequential kinetic scheme that best accounts for all of the kinetic data is shown. The steps targeted by the transcription factors RbpA (red), RbpA$^{CD-BL-SID}$ (green), or RbpA$^{BL-SID}$ (blue) are denoted. Arrows pointing at the relevant parameter indicate an increase in that parameter in the presence of the factor (compared to *Mbo* holo alone) by the fold-amount shown below.

quantitate the apparent slow growth phenotype, we determined the doubling times for each of the strains during exponential phase. Whereas RbpA$^{wt}$ doubled at 2.6(±0.16) hr, RbpA$^{CD-BL-SID}$ doubled slightly slower at 3.13(±0.16) hr and RbpA$^{BL-SID}$ and RbpA$^{R79A}$ doubled even more slowly at 4.1 (±0.21) hr and 4.6(±0.96) hr, respectively. In addition to growth rate, we monitored cell morphology using light microscopy. RbpA$^{wt}$ and RbpA$^{CD-BL-SID}$ were indistinguishable with regard to cell shape, membrane and nucleoid staining. By contrast, cells expressing RbpA$^{BL-SID}$ showed an array of morphological changes. These included filamentous cells, small cells, aberrant membrane staining, nucleoid condensation, and anucleate ghosts (*Figure 5A*). We scored nucleoid morphology as either condensed or diffuse in RbpA$^{WT}$ (n = 208) and RbpA$^{BL-SID}$ (n = 248) and observed that condensed nucleoids were present in 2.4% of RbpA$^{WT}$ cells, whereas 25% of RbpA$^{BL-SID}$ nucleoids were condensed (*Figure 5A*). RbpA$^{R79A}$ cells, despite their similarly slow growth rate, did not display

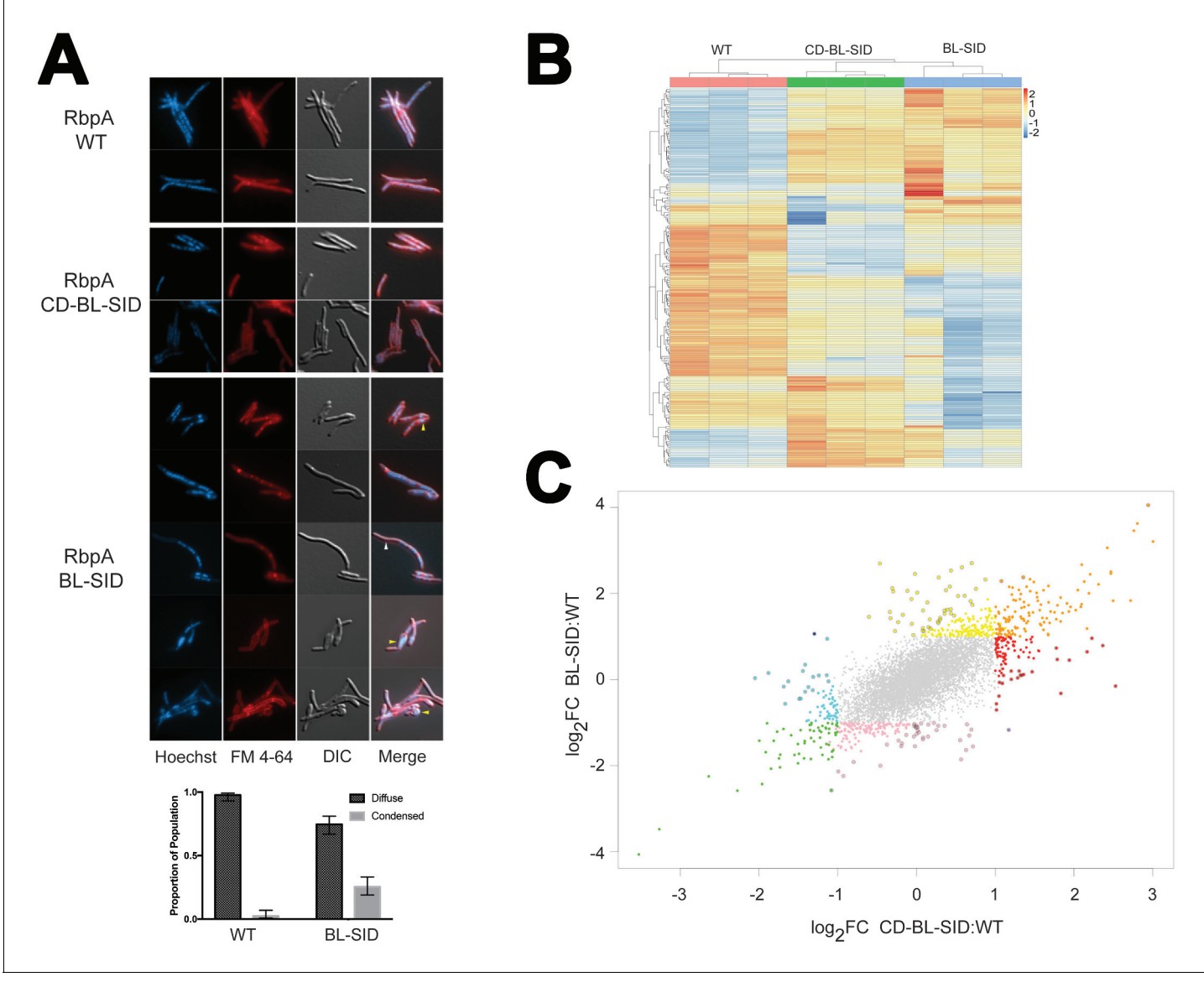

**Figure 5.** In vivo functions of RbpA NTT and CD. (A) Morphologic effects of RbpA truncations. *Msm* expressing either RbpA[wt], RbpA[CD-BL-SID], or RbpA[BL-SID] were stained with Hoechst (DNA) and FM 4–64 (membranes), and viewed by both fluorescence illumination and DIC imaging. Representative images are shown. The yellow arrowheads indicate aberrant cell morphology with condensed nucleoid in the RbpA[BL-SID] strain. The white arrowhead indicates an anucleate filament. Quantitation of nucleoid morphology in WT (n = 208) and RbpA[BL-SID] (n = 248) cells is graphed below the image as described in the text. Error bars are 99% confidence interval and p<0.0001 by chi squared test. (B) Unsupervised clustering of RNA-seq gene expression data from triplicate RNA samples from cells expressing RbpA[wt], RbpA[CD-BL-SID], or RbpA[BL-SID]. The cluster was generated from the 200 genes with the greatest variance between strains. The strains are clustered across the top of the heat map and genes clustered to the left of the heat map. (C) Distinct gene expression signatures associated with RbpA domains. Scatterplot of gene expression comparing the $\log_2$ fold change of RbpA[CD-BL-SID] (X axis), to RbpA[BL-SID] (Y axis) each compared to wild type cells. All colored points represent genes with statistically significant differences in RNA level compared to WT (adjusted p<0.01) with a fold change of >2 ($\log_2$ FC of >1 or <-1). The genes are classified by color according to their expression pattern as follows: orange (overexpressed in both strains), green (underexpressed in both strains), yellow (overexpressed in BL-SID but not CD-BL-SID), pink (underexpressed in BL-SID but not CD-BL-SID), red (overexpressed in CD-BL-SID but not BL-SID), cyan (underexpressed in CD-BL-SID but not BL-SID). Points with dark outline indicate genes in which the difference in fold change between the two strains is two fold or greater. The single dark blue and purple points represent genes in which the difference between strains is >2 fold but in opposite direction. See *Supplementary file 7* for gene lists corresponding to each color class.

morphologic changes compared to WT (data not shown). These findings indicate that, although the essential in vivo function of RbpA can be supplied without the NTT-CD, loss of the CD confers a pleiotropic phenotype of impaired viability and aberrant cell division, likely indicating broad effects on transcription.

### Differential effects of RbpA domains on gene expression in vivo

The viability of both the RbpA$^{CD-BL-SID}$ and RbpA$^{BL-SID}$ strains afforded an opportunity to test the effects of loss of these RbpA domains on gene expression. We executed RNA-sequencing on the RbpA$^{CD-BL-SID}$ and RbpA$^{BL-SID}$ strains compared to cells with RbpA$^{wt}$. Comparative analyses of the differentially expressed genes between RbpA$^{wt}$/RbpA$^{CD-BL-SID}$ and RbpA$^{wt}$/RbpA$^{BL-SID}$ revealed several patterns. First, despite the complete complementation by RbpA$^{CD-BL-SID}$ for viability, growth, and cell morphology, loss of the NTT had clear effects on gene expression that were distinct from RbpA$^{BL-SID}$, including both overexpressed and underexpressed genes (see red and blue dots, *Figure 5C*). These results clearly indicate that the NTT is functional in vivo and may be required for regulation of specific promoters. The RbpA$^{SID-BL}$ strain had an even more globally perturbed pattern of gene expression (*Figure 5B–C*), consistent with its pleiotropic phenotype (*Figure 5A*). Comparison between the strains indicated a core set of genes either up or downregulated by both RbpA alleles (*Figure 5B–C*; orange and green points, *Supplementary file 7*). Taken together, these data define distinct in vivo transcriptional functions for the NTT and CD, as well as a core set of genes that are dysregulated when either the NTT or CD is lost.

## Discussion

CarD and RbpA are key components of mycobacterial TICs. CarD is found at almost all σ$^A$ promoters in vivo (*Srivastava et al., 2013*), and available evidence points to RbpA being present at all σ$^A$ promoters as well. Both factors are essential in the major human pathogen *Mtb*. Our previous analyses of a thermus CarD/TIC showed that CarD stabilizes RPo by the conserved Trp wedging mechanism (*Bae et al., 2015a*; *Davis et al., 2015*; *Srivastava et al., 2013*). Here, we used in vitro and in vivo analyses to examine the mechanism for RbpA and determine how RbpA and CarD function together.

### Structure

The *Msm* RbpA/TIC structure reveals that RbpA participates in a complex network of interactions (*Figure 1*). The RbpA$^{CD}$ interacts with the β'zipper, which interacts with σ$^A_3$, both of which contact promoter DNA (*Figure 1B*). The RbpA$^{CD}$ also interacts with the ZBD, which interacts with σ$^A_4$, which in turn recognizes the −35 promoter element. The RbpA$^{SID}$ interacts with σ$^A_2$, which interacts with the −10 element. The RbpA/RNAP protein/protein interactions position RbpA$^{BL}$ residue R79 (and to a lesser extend K74 and k76) to interact with the DNA phosphate backbone just upstream of the −10 element (*Figure 1B*). Thus, RbpA enhances an intricate network of interactions between the ZBD, the zipper, σ$^A_2$, σ$^A_3$, σ$^A_4$, and promoter DNA.

### Pathway of RPo formation

We adapted a previously developed fluorescence assay (*Ko and Heyduk, 2014*; *Rammohan et al., 2015*) to monitor the kinetics of RPo formation and to reveal the steps that RbpA and CarD target to stimulate RPo formation (*Figures 3F* and *4D*). The assay is real-time, relatively non-perturbing (*Ko and Heyduk, 2014*), and highly versatile; it can be used to study the initiation process on virtually any promoter with any bacterial holo, and can be used to study the mechanistic details of extrinsic factors and small molecules that modulate bacterial transcription initiation, which have been estimated to number in the hundreds (*Ishihama, 2000*).

The kinetic mechanism of RPo formation by *Eco* holo has been extensively studied on four different promoters (*Buc and McClure, 1985*; *Rogozina et al., 2009*; *Rutherford et al., 2009*; *Saecker et al., 2011*; *Sclavi et al., 2005*). In each case, a sequential, three-step kinetic mechanism has been proposed (*Equation 1*), and this mechanism best explains the kinetic data for mycobacterial RNAP as well.

Crystal structures of the thermus and *Eco* RPo are available (*Bae et al., 2015b*; *Zuo and Steitz, 2015*), but structural information for RP1 and RP2 is limited to footprinting (*Saecker et al., 2011*).

These analyses have led to structural models of the kinetically significant intermediates schematized in *Figure 6A*. It is useful to discuss our kinetic results in light of these models (*Figure 6A*) and explicit structural models of the *Msm* TIC containing RbpA and CarD (*Figure 6B*).

## RP1

Footprinting of the first intermediate (called RP1 here) on several promoters indicates that the DNA is 'closed' (no $KMnO_4$ reactivity) and has not entered the active site cleft (*Cowing et al., 1989*; *Kovacic, 1987*; *Rogozina et al., 2009*; *Rutherford et al., 2009*; *Schickor et al., 1990*; *Sclavi et al., 2005*), leading to models like the one shown for RP1 (*Figure 6A*). This early intermediate has often been denoted 'RPc'. Several lines of evidence presented here indicate that RP1 represents an initial encounter complex between holo and duplex promoter DNA. First, the rate of forming RP1 appears diffusion-limited (~$10^8$ $M^{-1}s^{-1}$) for the two most active samples (*Eco* holo and *Mbo* holo+RbpA +CarD), ruling out large conformational changes in the RNAP or the DNA. Second, neither CarD nor RbpA affect $k_1$, $k_{-1}$, or $K_1$ significantly on either AP3 or VapB (*Table 1*, *Supplementary file 8*). Third, values of $k_1$, $k_{-1}$, and $K_1$ for AP3 (with a nearly consensus −35 element; *Figure 2—figure supplement 1A*) and for VapB (lacking a -35 element; *Figure 2—figure supplement 1A*) do not differ,

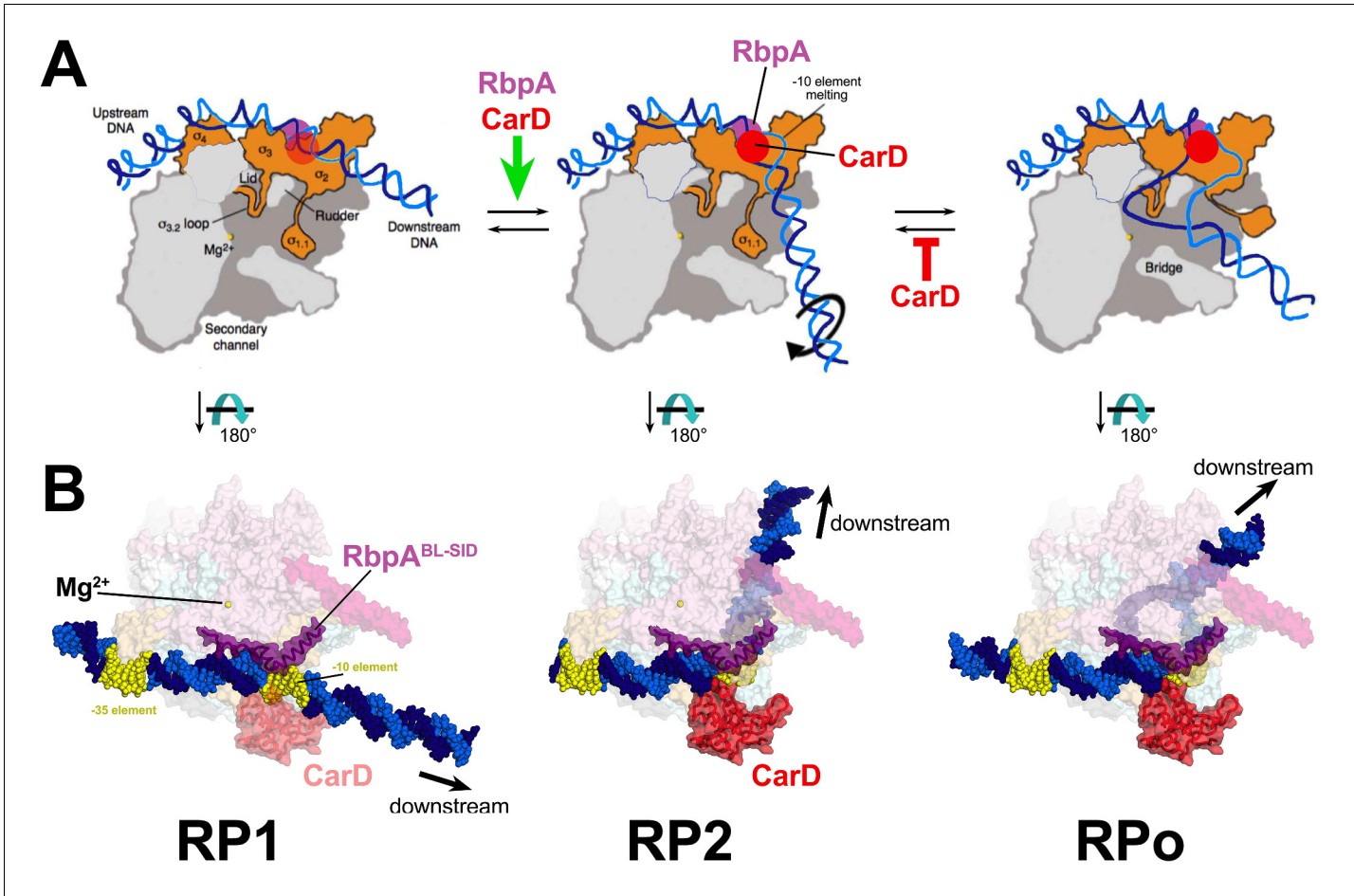

**Figure 6.** Structural transitions during the steps of RPo formation. (A) Schematic, cross-sectional views of the RNAP holo (catalytic $Mg^{2+}$, yellow sphere) and the promoter DNA (t-strand, dark blue; nt-strand, blue). RP1 and RP2 represent hypothetical models (*Saecker et al., 2011*). Crystal structures of RPo are available (*Bae et al., 2015b*; *Zuo and Steitz, 2015*). The important functional interaction of RbpA and CarD with the promoter DNA are schematically illustrated (RbpA, purple dot; CarD, red dot), based on the *Msm* RbpA/TIC structure (*Figure 1*). (B) Explicit structural models of the *Msm* RbpA/CarD/TIC with promoter DNA modeled as in the hypothetical models in (A). The RNAP is shown as a transparent molecular surface (color-coded as in *Figure 1*). RbpA and the modeled CarD (*Figure 2D*) are shown as transparent molecular surfaces with the backbone ribbon also shown. The −35 and −10 elements are colored yellow.

suggesting that sequence-specific readout of the of the −35 element by $\sigma_4$ occurs in subsequent steps (*Campbell et al., 2002*).

The −10 element sequence is recognized only in its single-stranded form (*Feklistov and Darst (2011)*, but RP1 DNA is closed. How is initial promoter recognition achieved if DNA sequence-specific interactions upstream of the −10 element are not critical for formation of RP1? Non-sequence-specific protein/DNA interactions presumably orient holo with respect to the duplex DNA in a way that promotes subsequent steps leading to RPo formation. In particular, we suggest that in RP1, an important component of promoter recognition by holo may be mediated through distinct conformational characteristics of promoter DNA (*Feklistov, 2013*). Indeed, the chemical probe 1,10-phenanthroline copper reacts uniquely with the −10 element region of several promoters in the absence of any proteins, indicating special conformational characteristics (*Spassky et al., 1988*; *Spassky and Sigman, 1985*).

While *Rammohan et al. (2015)* suggested that CarD destabilizes an initial, RPc-like complex on AP3, we find that CarD has no effect on the stability of RP1 and stabilizes both RP2 and RPo (*Figure 3F*). We suggest that their proposal is inconsistent with our results due the following: The inherent difficulty of trying to deduce mechanisms from multi-exponential fits to the data, working at non-saturating concentrations of CarD where mixed populations of CarD bound and unbound species without distinct fluorescent signals exist, and finally interpreting these results within the framework of a two-step kinetic mechanism instead of the three-step mechanism shown here to be necessary to account for the data (Appendix). This latter simplification can lead to erroneous interpretations (*Tsodikov and Record, 1999*). Our analysis yielded estimates for the forward and reverse rate constants for the three kinetic steps (*Table 1*). The promoter half-lives calculated from these rate constants on the AP3 promoter matched the independently, experimentally determined values (*Figure 2C*, *Supplementary file 4*), supporting the validity of our approach.

## RP2

The conversion of RP1 to the next significant intermediate extends protection of DNA from cleavage reagents further downstream to +10 and beyond (*Rutherford et al., 2009*; *Saecker et al., 2011*; *Sclavi et al., 2005*). Models for this intermediate posit a sharp bend of the DNA at the −10 element (RP2, *Figure 6A*), positioning the downstream duplex DNA across the entrance to the RNAP active site cleft (*Saecker et al., 2011*). While this intermediate appears closed on some promoters (*Rutherford et al., 2009*; *Saecker et al., 2011*), partial transcription bubble formation has been observed on T7A1 (*Rogozina et al., 2009*). Limited transcription bubble formation could be hidden from $KMnO_4$ footprinting because of interactions of unstacked bases with the RNAP (*Bae et al., 2015b*), and current models suggest that at least one base (the highly conserved −11A of the −10 element nt-strand; *Feklistov and Darst, 2011*) is flipped out of the DNA duplex in this intermediate, facilitating the kink in the DNA (*Saecker et al., 2011*; *Werel et al., 1991*; *Figure 6A*).

While the −35 element does not appear to be important for the formation of RP1, it is very important for the progression of RP1 to RP2. Indeed the most striking kinetic difference between VapB (lacking a −35 element) and AP3 (with a −35 element) is in the RP1 -> RP2 transition ($k_2$, $k_{-2}$, $K_2$; *Supplementary file 6*): the VapB $k_2$ is reduced nearly 50-fold (*Mbo* holo) and 230-fold (*Eco* holo) compared to AP3. The simple interpretation of these large effects is that DNA sequence-specific, $\sigma_4$/−35 element interactions (*Campbell et al., 2002*) are established as RP2 forms, providing part of the favorable binding free energy to drive the RP1 -> RP2 transition. Previous work has shown that stabilizing upstream $\sigma_4$/DNA interactions can stimulate isomerization to RPo (*Dove et al., 2000*).

Interactions upstream of the −10 element may play a key structural role in facilitating the −11/−12 kink in the DNA (*Figure 6*) by anchoring the lateral position of RNAP on the promoter DNA and/or by constraining the twist of the DNA. These upstream interactions may effectively position the −10 element with respect to the pocket on $\sigma^A_2$ that captures the flipped out −11A (*Feklistov and Darst, 2011*). On both AP3 and VapB, RbpA affects only $k_2$, with the largest effect on VapB (*Table 1*; *Supplementary file 8*; *Figures 3F* and *4D*), suggesting that the interaction of RbpA-R79 with the nt-strand −13 phosphate (*Figure 1B*) fulfills this anchoring role when $\sigma_4$/−35 element interactions cannot form. We note the majority of *Mtb* promoters appear to lack a −35 region (*Cortes et al., 2013*), suggesting why, in part, RpbA is required for cell viability.

In the case of *Mbo* holo on AP3, CarD alone stimulates $k_2$ ~10 fold. In the model of RP1, the position of the CarD-CTD clashes with duplex DNA (*Figure 6A*; *Srivastava et al., 2013*; *Bae et al.,*

2015a). Favorable interactions of the CarD-CTD with promoter DNA require widening of the minor groove at the upstream edge of the −10 element (from about −14 to −10), which is coupled with initial opening of the −10 element (*Bae et al., 2015a*) and bending of the DNA, sending it towards the active site channel (*Figure 6B*). While flexibility between the CarD-RID (interacting with the RNAP β-lobe 1) and the CarD-CTD (*Gulten and Sacchettini, 2013*) combined with RNAP clamp opening may alleviate any clash in RP1, the RP1 model illustrates how the favored position of the CarD-CTD could drive RP2 formation by stabilizing the initial formation of the upstream duplex/single stranded junction (*Figure 6B*).

The 2-AP experiments (*Figure 3D–E*) indicate that opening of the downstream edge of the transcription bubble does not occur at a rate comparable to the formation of RP2 but at the slower rate of RPo formation. We conclude that *Mbo* holo on AP3 forms the transcription bubble in a two-stage process, with initial opening of the −10 element in RP2. Further downstream opening then creates the transcription bubble in RPo (*Figure 6*).

CarD and RbpA together have a cooperative effect on the RP1 -> RP2 transition, and both factors interact with the promoter DNA at or near the upstream edge of the −10 element (*Figures 2D* and *6B*). A CarD/RbpA protein/protein interaction in the context of the TIC could explain this cooperativity, but this seems unlikely on the basis of the structural modeling (*Figure 2D*). The RbpA$^{NTT}$ is not accounted for in our structure, but deletion of the NTT (and the NTT-CD) does not abrogate the CarD/RbpA cooperativity (*Figure 2B*). Rather, RbpA and CarD affect the same step (RP1 -> RP2 transition), but do so non-competitively and in different ways. In this scheme, initial opening of the −10 element and DNA bending are coupled. Through its anchoring role, RbpA stimulates formation of RP2 through facilitating bending, which is coupled with initial opening of the −10 element, facilitating CarD function. Through its wedging role, CarD stabilizes the initial −10 opening, which is coupled to bending, facilitating RbpA function.

## RPo

The RP2 -> RPo transition opens the DNA duplex downstream to the start site (*Figure 3D*) and loads the t-strand DNA into the RNAP active site (*Figure 6C*). The step involving full transcription bubble formation on λP$_R$ is rate-limiting in the forward direction, and the same step is rate-limiting in the reverse direction (*Saecker et al., 2011*). On *rrnB* P1, stabilization of the full transcription bubble requires ATP and CTP, the first two nucleotides of the transcript (*Rutherford et al., 2009*), indicating that expansion of the transcription bubble downstream to the start site is energetically difficult in this case as well. On AP3, opening of the full transcription bubble occurs during the RP2 -> RPo transition (*Figure 3D–E*), and this transition is rate-limiting in both directions (*Figure 3—figure supplement 1F*).

## RbpA function in vivo

Our analyses of RbpA function in vivo confirm that RbpA, like CarD, is essential for viability. Loss of the NTT or NTT-CD, despite relatively minor biochemical effects in vitro on the tested promoters (*Figures 2B–C* and *4C–D*; *Supplementary file 8*), has clear effects on gene expression in vivo (*Figure 5B–C*). Although all of these effects cannot be attributed directly to RbpA function at present, some of the changes are likely direct effects of the RbpA truncations at specific promoters. Similarly, we detected a core set of genes that are dysregulated in both RbpA alleles, suggesting that some of these may be direct RbpA targets, raising the possibility that the truncated RbpA segments may interact with additional, as yet unidentified, transcription regulators.

In our in vitro analyses, RbpA$^{R79A}$ by itself appeared to have lost all RbpA function in activating transcription and counteracted the activity of CarD when both factors were present (*Figure 2B*, *Supplementary file 8*). Therefore, our finding that RbpA$^{R79A}$ supported viability was surprising. However, our results clearly show that the in vivo role of RbpA is very complex (*Figure 5*), and in this work we have only examined two *bona fide* mycobacterial promoters, AP3 and VapB. It seems likely that other promoters are regulated by RbpA$^{R79A}$ *in vivo*, possibly through the action of additional conserved basic residues on the BL such as K76 (*Figure 1—figure supplement 1D*). Motif searches, direct assays of RbpA genome wide promoter binding, and identification of potential protein binding partners will be needed to begin to unravel the specific RbpA regulated gene sets and/or

interaction partners that mediate its essential function in vivo and the rules that may determine the differential effects of RbpA at specific promoters.

## Conclusions

The work presented here provides an unprecedented structural framework for understanding myco-bacterial transcription. With our in vitro functional analysis and in combination with our previous analyses of CarD structure and function (*Bae et al., 2015a*; *Davis et al., 2015*; *Srivastava et al., 2013*), we propose the following model (*Figure 6B*): RbpA-R79 interaction with duplex DNA upstream of the −10 provides an anchoring role that facilitates the bending (RP2) and subsequent formation of the full transcription bubble (RPo). Initial distortions of the DNA in RP2 are compatible with CarD to wedge its conserved Trp residue into the splayed minor groove, which not only contributes to formation of the RP2 intermediate, but also serves as a barrier for bubble collapse once RPo has formed. Our in vivo analyses point to a complex role for the various structural elements of RbpA in promoter specific regulation and possible interactions with additional factors. The role of RbpA in facilitating the formation of RP2 is particularly important at promoters lacking sequence elements upstream of the −10 element, which represents the majority of *Mtb* promoters (*Cortes et al., 2013*), possibly explaining why RbpA is essential and spotlighting the importance of studying a diversity of organisms in efforts to characterize 'conserved' processes such as transcription.

## Materials and methods

Standard procedures were used to manipulate recombinant DNA and to transform *Eco. Msm* strains were derivatives of $mc^2155$ (*Snapper et al., 1990*). *Msm* was transformed by electroporation (2500 V, 2.5 μF, 1000 Ω). All *Msm* strains were cultured in Lbsmeg (LB with 0.5% glycerol, 0.5% dextrose, and 0.05% Tween$_{80}$). Antibiotic concentrations used for selection of *Msm* strains were as follows: kanamycin 20 μg/ml, hygromycin 50 μg/ml, streptomycin 20 μg/ml. All *Msm* strains, plasmids with relevant features, and primers used in generating the *Msm* strains are listed in *Supplementary files 9* and *10*.

### Protein expression and purification

#### Msm RbpA/$\sigma^A$

*Msm* pET-SUMO $\sigma^A$ (received from C. Stallings) and pet21C-RbpA (cloned from *Msm* mc2155 genomic DNA) were co-transformed into BL21(DE3) *Eco* cells. Protein expression was induced at 30°C for 3 hr with 0.5 mM IPTG at O.D.$_{600nm}$ = 0.6 and purified by Ni$^{2+}$-affinity chromatography (HiTrap IMAC HP, GE Healthcare Life Sciences, Pittsburgh, PA) as described (*Hubin et al., 2015*). Following elution, the complex was dialyzed overnight into 20 mM Tris-HCl, pH 8, 0.5 M NaCl, 5 mM imidazole, 5% (v/v) glycerol, 0.5 mM $\beta$-mercaptoethanol, and the His$_{10}$-SUMO tag was cleaved by incubation with ULP1 protease overnight at a ratio of 1/25 (ULP1 mass/cleavage target mass). The cleaved complex was loaded onto a second Ni$^{2+}$-affinity column and was retrieved from the flow-through and low-imidazole washes. The complex was loaded directly on a size exclusion column (SuperDex-200 16/16, GE Healthcare Life Sciences) equilibrated with 20 mM Tris-HCl, pH 8, 0.5 M NaCl, 5% glycerol (v/v), 1 mM DTT, 1 mM EDTA. The sample was concentrated to 15 mg/ml by centrifugal filtration and stored at −80°C.

#### Msm RNAP – rpoC-ppx-His$_{10}$ construction

To construct an *Msm* strain for endogenous RNAP purification, we inserted a PreScission Protease (ppx; GE Healthcare Life Sciences) and His$_{10}$-tag (ppx-His$_{10}$-tag) at the 3' end of the *rpoC* gene by single crossover integration of pAJF527, which carries 554 bp of the 3' end of *rpoC* fused to the ppx cleavage sequence (LEVLFQGP) and His$_{10}$ affinity tag. pAJF527 was transformed into MC2155 and transformants were selected with hygromycin. Transformants were screened for integration at *rpoC* by PCR amplification using primers that anneal to 79 bp upstream of the 3' flank (oAF799) and within the integrating vector downstream of the His$_{10}$-tag (oAF021). RNAP $\beta$'-ppx-His$_{10}$ expression was confirmed by immunoblot using HisProbe (ThermoScientific Pierce, Waltham, MA), which detected a protein species at approximately 150 kDa. The resulting strain (MGM6029) showed no growth

defects under standard laboratory conditions, indicating that the tagged RNAP $\beta'$ subunit is functional (as it is the only copy in the cell).

## *Msm* RNAP purification

Endogenous *Msm* RNAP was purified from the *Msm* mc2155 strain expressing a native chromosomal copy of *rpoC* with a C-terminal ppx-His$_{10}$-tag. *Msm* cells were grown at the Bioexpression and Fermentation Facility at the University of Georgia. Cells were lysed by continuous flow French press (Avestin) in 50 mM Tris-HCl, pH 8, 1 mM EDTA, 5% (v/v) glycerol, 5 mM DTT, 1 mM protease inhibitor cocktail, 1 mM phenylmethylsulfonyl fluoride, and core RNAP was precipitated from the cleared lysate by polyethyleneimine (PEI) precipation (0.35%). The PEI pellet was washed four times with 10 mM Tris-HCl, pH 8, 0.5 M NaCl, 0.1 mM EDTA, 5 mM DTT, 5% (v/v) glycerol, then eluted four times with the same buffer but with 1 M NaCl. Protein was precipitated overnight with 35% (w/v) ammonium sulfate and resuspended in 20 mM Tris-HCl, pH 8, 5% (v/v) glycerol, 1 M NaCl, 1 mM $\beta$-mercaptoethanol. Protein was loaded on a Ni$^{2+}$-affinity column (HiTrap IMAC HP, GE Healthcare Life Sciences) and eluted in 20 mM Tris-HCl, pH 8, 5% (v/v) glycerol, 0.5 M NaCl, 0.25 M imidazole. Protein was diluted in 10 mM Tris-HCl, pH 8, 5% (v/v) glycerol, 0.1 mM EDTA, 5 mM DTT to a final salt concentration of 0.1 M NaCl, loaded on a Biorex (BioRad, Hercules, CA) ion exchange column, and eluted with a gradient from 0.1 M – 0.8 M NaCl. A 5X molar excess of the purified *Msm* RbpA/$\sigma^A$ complex was added to the RNAP and the resulting complex was purified by size exclusion chromatography (Superdex-200, GE Healthcare Life Sciences) in 20 mM Tris-HCl, pH 8, 5% (v/v) glycerol, 0.5 M NaCl, 0.25 M imidazole. The purified complex was dialyzed into 20 mM Tris-HCl, pH 8, 100 mM K-glutamate, 10 mM MgCl$_2$, 1 mM DTT, concentrated by centrifugal filtration to ~15 mg/ml, and stored at −80°C.

## *Eco/Mtb/Mbo proteins*

*Eco* core RNAP, *Eco* σ$^{70}$, *Mtb/Mbo* σ$^A$, RbpA, RbpA$^{R79A}$, CarD, and recombinant *Mbo* core RNAP were expressed and purified as described (*Davis et al., 2015*; *Hubin et al., 2015*). *Mtb/Mbo* RbpA truncations (*Figure 2A*; *Hubin et al., 2015*) were expressed in BL21(DE3) *Eco* cells from a pET-SUMO vector and purified using the same procedure as for RbpA.

## Crystallization of *Msm* RbpA/TIC

The upstream-fork promoter fragment (T10; *Figure 1—figure supplement 1A*) was assembled from synthetic oligos (Integrated DNA Technologies, Coralville, IA) by annealing in 10 mM Tris-HCl, pH 8, 1 mM EDTA, 0.2 M NaCl. Aliquots were stored at −20°C. The purified *Msm* RbpA/σ$^A$-holo was mixed in a 1:1 molar ratio with annealed T10 upstream fork DNA to generate the *Msm* RbpA/TIC. Crystals were grown by hanging drop vapor diffusion by mixing 1 μL of *Msm* RbpA/TIC solution (11 mg/mL protein) with 1 μL of crystallization solution [0.1 M Bis-Tris, pH 6.0, 0.2 M LiSO$_4$, 16% (w/v) polyethylene glycol 3350, 2.5% (v/v) ethylene glycol] and incubating over a well containing crystallization solution at 22°C. Crystals grew to full size in approximately 10 days, with the largest crystals (very rare) having dimensions ~200 × 200 x 1000 μm. The crystals were cryo-protected by step-wise transfer (three steps) into 0.1 M Bis-Tris, pH 6.0, 0.2 M LiSO$_4$, 22% (w/v) polyethylene glycol 3350, 20% (v/v) ethylene glycol and flash frozen by plunging into liquid nitrogen.

## Data collection, structure determination, and refinement

X-ray diffraction data were collected at the Argonne National Laboratory Advanced Photon Source (APS) NE-CAT beamline 24-ID-E. Structural biology software was accessed through the SBGrid consortium (*Morin et al., 2013*). Data were integrated and scaled using HKL2000 (*Otwinowski and Minor, 1997*).

An initial electron density map was calculated by molecular replacement using Phaser (*McCoy et al., 2007*) from a starting model generated from *Thermus aquaticus* RNAP holo (PDB ID 4XLP; *Bae et al., 2015b*) but with σ$^A_2$ deleted and replaced with *Mtb* σ$^A_2$ (from PDB ID 4X8K; *Hubin et al., 2015*). One RbpA/TIC complex was clearly identified in the asymmetric unit. The model was first improved using rigid body refinement of 20 individual mobile domains using PHENIX (*Adams et al., 2010*). The resulting model was improved by iterative cycles of manual building with COOT (*Emsley and Cowtan, 2004*) and refinement with PHENIX (*Adams et al., 2010*). The

PDBePISA server (http://www.ebi.ac.uk/pdbe/pisa/) was used to calculate intermolecular buried surface areas.

## In vitro transcription assays

In vitro abortive initiation transcription assays and promoter lifetime assays (*Figure 2*) were performed at 37°C as described (*Davis et al., 2015*). AP3[anti-35] promoter was prepared using PCR amplification on a synthesized template (Integrated DNA Technologies). Assays with the AP3[anti-35] promoter were performed in KGlu assay buffer (10 mM Tris-HCl, pH 8.0, 100 mM K-glutamate, 10 mM $MgCl_2$, 0.1 mM EDTA, 0.1 mM DTT, 50 μg/mL BSA). All other assays were performed in KCl assay buffer (10 mM Tris-HCl, pH 8.0, 50 mM KCl, 10 mM $MgCl_2$, 0.1 mM EDTA, 0.1 mM DTT, 50 μg-/mL BSA).

*Mbo* core (50 nM) was incubated with $\sigma^A$ (250 nM) for 5 min at 37°C to form holo and, when used, CarD and or RbpA or RbpA derivatives (all 2 μM) were added and incubated for an additional 5 min. DNA template (10 nM) was then added and incubated with the protein mixture for 15 min to form RPo. Abortive initiation assays were initiated on VapB with ApU dinucleotide primer (250 μM; Trilink Biotechnologies, San Diego, CA), [$\alpha$-$^{32}$P]GTP (1.25 μCi; Perkin Elmer Life Sciences, Waltham, MA), and unlabeled GTP (50 μM; GE Healthcare Life Sciences). Abortive transcription for abortive initiation and promoter lifetime assays were initiated on AP3 with GpU dinucleotide primer (250 μM; Trilink Biotechnologies), [$\alpha$-$^{32}$P]UTP (1.25 μCi; Perkin Elmer Life Sciences), and unlabeled UTP (50 μM; GE Healthcare Life Sciences). Competitor DNA for the promoter lifetime assays was full-con bubble promoter trap DNA (*Figure 3—figure supplement 1A* from *Davis et al., 2015*). Transcription products were visualized by polyacrylamide gel electrophoresis (23%) followed by phosphorimagery and quantitation using Image J.

## Stopped-flow kinetics with Cy3-DNA

### Preparation of DNA.

Cy3-AP3 promoter DNA (*Figure 3—figure supplement 1A*) was prepared using Cy3-amido-dT modified DNA (Cy3-AP3-DNA1):

5′CATCTATGGATGACCGAACCTGGTCTTGACTCCATTGCCGGATTTGTATTAGACTGGCAGGG TTG/ICY3N/TG3′ from Integrated DNA Technologies. Cy3-AP3-DNA1 (0.25 μM) was mixed with AP3-DNA2 (0.25 μM; Integrated DNA Technologies):

5′TTCTGAGTTCGGCATGGGGTCAGGTGGGACCCAAGCTTCCGCTTCGGGGCAACCCTGCCAG TCTAATAC3′

The annealed oligos were extended using *Taq* DNA polymerase with 20 cycles (30 s each) of heating to 95°C, annealing at 48°C, and extending at 72°C. The resulting product was visualized on a non-denaturing 6% polyacrylamide gel stained with Gel Red (Biotium, Fremont, CA) to verify a single product, and then purified using PCR clean up (Qiagen, Hilden, Germany).

Cy3-VapB promoter DNA (*Figure 3—figure supplement 1A*) was prepared as above but with the following starting synthetic oligos (Integrated DNA Technologies):

Cy3-VapB-DNA1:

5′GGCTGAATCGCCGCCCGCCGCGGTGCCGCCCGGGCCGCACATTGTGATGTATGATATGGTG TA/ICY3N/G3′

VapB-DNA2:

5′CGTGTAACACTACATACTATACCACATACTTCGCCTGGTTGTAGATGGAGCTGCTCCTCGTC TGCCGTTCGGAGCTGTT3′

### Stopped-flow assays – forward kinetics

To monitor the kinetics of RPo formation, proteins (RNAP and transcription factors if present, see *Supplementary files 3* and *5* for assay conditions) were loaded in one syringe of a stopped-flow instrument (AutoSF-120, KinTek Corporation, Snowshoe, PA) and Cy3-DNA was loaded into the other. After rapid mixing at 37°C, the final solution conditions were 10 mM Tris-HCl, pH 8, 50 mM K-glutamate, 20 mM NaCl, 10 mM $MgCl_2$, 1 mM DTT. Cy3 fluorescence emission was monitored in real time with a 586/20 single-band bandpass filter (Semrock, Rochester, NY) with excitation at 515 nM.

## Stopped-flow assays – reverse kinetics

To monitor the kinetics of RPo dissociation for *Mbo* holo and *Mbo* holo+RbpA, RPo was preformed on Cy3-AP3 DNA by incubating at 37°C for 30 min prior to loading into one syringe of the stopped-flow instrument. The other syringe was loaded with full-con bubble promoter trap DNA (*Davis et al., 2015*). After rapid mixing at 37°C, the final solution conditions were *Mbo* holo (250 nM), RbpA (when present, 5 μM), Cy3-AP3 DNA (1 nM), full-con bubble promoter trap DNA (1 μM). Cy3 fluorescence emission was monitored in real time as above. For each sample, three independent 'shots' were averaged together (*Figure 3—figure supplement 1D–F*).

## Stopped-flow assays with 2-AP-AP3 DNA

The 2-AP-AP3 (bottom strand) DNA (*Figure 3—figure supplement 1G*) was obtained from Trilink Biotechnologies, 2-AP-AP3 (top strand) was obtained from Integrated DNA Technologies. The two synthetic oligos were annealed using the same procedure described above. To monitor the kinetics of RPo formation, *Mbo* holo was loaded in one syringe of the stopped-flow instrument, and 2-AP-AP3 DNA was loaded in the other. After rapid mixing at 37°C, the final solution conditions were *Mbo* holo (250 nM), 2-AP-AP3 DNA (10 nM). The 2-AP fluorescence emission was monitored in real time with a 357/44 nm BrightLine single-band bandpass filter (Semrock) with excitation at 315 nm. The curve shown in *Figure 3D* was the result of averaging six 'shots'.

Data analysis is described in the text and in the Appendix.

## Δ*rbpA* construction

Gene deletion was made by homologous recombination and double negative selection (*Barkan et al., 2011*). Briefly, approximately 500 bp upstream of the coding sequence and 500 bp downstream of the coding sequence of *rbpA* (MSmeg_3858) were cloned into the suicide plasmid pAJF067 containing a hygromycin cassette for positive selection of integration and two markers for negative selection, *sacB* and *galK*. After transformation of *Msm* with this plasmid, hygromycin resistant colonies resulting from a single crossover event were verified as intermediates by PCR. A merodiploid was then generated by adding a second copy of *rbpA* at *attB*(L5) by transformation with pAJF685 and subsequently a second crossover was selected to generate Δ*rbpA* by counterselecting on sucrose and 2-deoxygalactose containing agar media after passaging in the absence of hygromycin. Deletion mutants were verified by PCR. To assess if *rbpA* was essential, allele swapping was performed for the *attB* integrated copy of *rbpA* using marker exchange for either pMV306kn (empty vector), pAJF672 (encoding full length RbpA), pAJF679 (RbpA(28-114aa) or pAJF680 (RbpA(72-114aa). No transformants were recovered from the empty vector control transformations, confirming the essentiality of *rbpA*.

## RbpA growth curves

All growth curves were performed in LBsmeg (no antibiotics) at 37°C. Triplicate cultures were grown overnight to mid logarithmic phase and then diluted back. After 15 hr of growth when cultures reached an OD600 of 0.1, measurements were taken every 2 hr until the cells reached an OD600 of 0.6. Doubling times were then calculated.

## Microscopy

All images were acquired using a Zeiss (Oberkochen, Germany) Axio Observer Z1 microscope equipped with, Colibri.2 and Illuminator HXP 120 C light sources, a Hamamatsu (Hamamatsu, Japan) ORCA-Flash4.0 CMOS Camera and a Plan-Apochromat 100×/1.4 oil DIC objective. Zeiss Zen software was used for acquisition and image export. The following filter sets and light sources were used for imaging: Hoechst 33342 (49 HE, HXP 120 C) and FM 4–64 (20, HXP 120 C). For cell staining 100 μl of culture was used. A final concentration of 1 μg/ml FM 4–64 (Invitrogen, Carlsbad, CA) and/or 10 μg/ml Hoechst 33342 (Invitrogen) was added. Cells were collected by centrifugation at 5000 g for 1 min and resuspended in 50 μl of media. For time-lapse microscopy, cells were added to a 1.5% low melting point agarose PBS pad. Agarose was heated to 65°C and poured into a 17 × 28 mm geneframe (Thermoscientific, AB-0578) adhered to a 25 × 75 mm glass slide. A second slide was pressed down on top and the set-up was allowed to cool at room temperature for 10 min. 2–3 μl of

*Msm* culture was added to the pad and a No. 1.5 24 × 40 mm coverglass was sealed to the geneframe.

## RNA purification

Bacterial culture (50 mL) normalized to an $OD_{600}$ of 0.4 was cooled to 4°C and harvested by centrifugation. Pellets were resuspended in RNALater (Invitrogen) and stored overnight at 4°C. Pellets collected by centrifugation were washed in 1 ml of 10 mM Tris-HCl, pH 8.0. Pellets were then resuspended in 100 µl $TE_{80}$ with 1 mg/ml lysozyme and disrupted by bead beating with a Fast-Prep120 2 times at 5.0 m/s for 25 s. This lysate was used for RNA purification with a GeneJet RNA purification kit (Thermoscientific) following the manufacturer's protocol. RNA was eluted in 85 µl elution buffer and then treated with DNase I (Thermoscientific) for 30 min at 37°C. GeneJet purification columns were used to clean RNA from DNaseI reactions.

## RNA sequencing and data analysis

Sample libraries were prepared for RNA-seq using the Ribo-Zero Magnetic Bacterial kit (Epicentre, Madison, WI) in connection with TruSeq Stranded Total RNA kit (Illumina, San Diego, CA). Paired-end reads ($10–20 \times 10^6$) were obtained for each sample replicate on an Illumina HiSeq 2500. Post-run demultiplexing and adapter removal were performed and fastq files were inspected using fastqc (*Andrews, 2010*). Trimmed fastq files were then aligned to the reference genome (M. smegmatis $MC^2155$; NC_008596.1) using bwa mem (*Li and Durbin, 2009*). Bam files were sorted and merged using samtools (*Li et al., 2009*) and gene counts were obtained using featureCounts from the Bioconductor Rsubread package (*Liao et al., 2014*). Differentially expressed genes were identified using the DESeq2 R package (*Love et al., 2014*) and subsequent analysis of gene expression was performed in R (*Kolde, 2015*; *R Core Team, 2016*).

## Accession numbers

The X-ray crystallographic coordinates and structure factor file have been deposited in the Protein Data Bank with accession ID 5TW1. The RNA-seq data have been deposited in the Gene Expression Omnibus (GEO) database with accession number GSE89773.

## Acknowledgements

We thank R Landick, and BT Nixon for helpful discussions, N Socci in the MSKCC bioinformatics core for assistance with RNAseq mapping. This work is based upon research conducted at the Northeastern Collaborative Access Team beamlines, which are funded by the NIGMS from the NIH (P41 GM103403). The Pilatus 6M detector on 24-ID-C beam line is funded by an NIH-ORIP HEI grant (S10 RR029205). This research used resources of the Advanced Photon Source, a US Department of Energy (DOE) Office of Science User Facility operated for the DOE Office of Science by Argonne National Laboratory under Contract No. DE-AC02-06CH11357. The use of the Rockefeller University Structural Biology Resource Center was made pssible by NIH/NCRR 1S10RR027037. This work was supported by NIH grant RO1 GM114450 to EAC and P30 CA008748.

## Additional information

### Funding

| Funder | Grant reference number | Author |
| --- | --- | --- |
| National Institutes of Health | RO1 GM114450 | Elizabeth A Campbell |
| National Institutes of Health | P30 CA008748 | Michael S Glickman |

The funders had no role in study design, data collection and interpretation, or the decision to submit the work for publication.

### Author contributions

EAH, Conceptualization, Investigation, Methodology, Writing—review and editing; AF, CX, Investigation, Methodology; JMB, Formal analysis, Methodology; RMS, Methodology, Writing—review and

editing; MSG, Conceptualization, Supervision, Funding acquisition, Writing—review and editing; SAD, Conceptualization, Formal analysis, Supervision, Validation, Methodology, Writing—original draft, Writing—review and editing; EAC, Conceptualization, Supervision, Funding acquisition, Investigation, Methodology, Writing—original draft, Writing—review and editing

**Author ORCIDs**
Michael S Glickman, http://orcid.org/0000-0001-7918-5164
Seth A Darst, http://orcid.org/0000-0002-8241-3153

# Additional files

## Supplementary files
• Supplementary file 1. Table showing sequence identity of *Msm*, *Mtb*, and *Mbo* transcription initiation proteins.

• Supplementary file 2. Table of crystallographic statistics for the *Msm* RbpA/TIC crystals.

• Supplementary file 3. Table of conditions for kinetic experiments with the Cy3-AP3 promoter.

• Supplementary file 4. Table of unconstrained kinetic parameters on the *Mtb* AP3 promoter.

• Supplementary file 5. Table of conditions for kinetic experiments with the Cy3-VapB promoter.

• Supplementary file 6. Table comparing kinetic parameters for *Mbo* holo and *Eco* holo on AP3 vs. VapB promoters.

• Supplementary file 7. Excel file listing transcripts identified from the RNA-seq data as well as gene lists corresopnding to the different colored datapoints in *Figure 5B*).

• Supplementary file 8. Table of kinetic parameters on the *Mtb* VapB promoter.

• Supplementary file 9. *Msm* strains and oligos used in this study. (A) Table of *Msm* strains used in this study. (B) Table of oligos used in generating *Msm* strains.

• Supplementary file 10. Table of plasmids used in generating *Msm strains*.

## Major datasets
The following datasets were generated:

| Author(s) | Year | Dataset title | Dataset URL | Database, license, and accessibility information |
|---|---|---|---|---|
| Hubin EA, Darst SA, Campbell EA | 2016 | Crystal structure of a Mycobacterium smegmatis transcription initiation complex with RbpA | http://www.rcsb.org/pdb/explore/explore.do?structureId=5tw1 | Publicly available at Protein Data Bank (accession no: 5TW1) |
| Fay A, Glickman MS | 2016 | Gene expression profile of M. smegmatis expressing RbpA truncations | https://www.ncbi.nlm.nih.gov/geo/query/acc.cgi?acc=GSE89773 | Publicly available at NCBI Gene Expression Omnibus (accession no: GSE89773) |

The following previously published datasets were used:

| Author(s) | Year | Dataset title | Dataset URL | Database, license, and accessibility information |
|---|---|---|---|---|
| Hubin EA, Flack JE, Tabib-Salazar A, Paget MS, Darst SA, Campbell EA | 2014 | *Mycobacterium tuberculosis* RbpA-SID in complex with SigmaA domain 2 | http://www.rcsb.org/pdb/explore/explore.do?structureId=4X8K | Publicly available at Protein Data Bank (accession no: 4X8K) |
| Bortoluzzi A, Muskett FW, Waters LC, Addis PW, Rieck B, Munder T, Schleier S, Forti F, Ghisotti D, Carr MD, O'Hare HM | 2013 | *Mycobacterium tuberculosis* RNA polymerase binding protein A (RbpA) and its interactions with sigma factors | http://www.rcsb.org/pdb/explore/explore.do?structureId=2M4V | Publicly available at Protein Data Bank (accession no: 2M4V) |
| Bae B, Darst SA | 2015 | Crystal structure of *T.aquaticus* transcription initiation complex with CarD containing bubble promoter and RNA | http://www.rcsb.org/pdb/explore/explore.do?structureId=4XLR | Publicly available at Protein Data Bank (accession no: 4XLR) |
| Bae B, Darst SA | 2015 | Crystal structure of *T.aquaticus* transcription initiation complex containing upstream fork promoter | http://www.rcsb.org/pdb/explore/explore.do?structureId=4XLP | Publicly available at Protein Data Bank (accession no: 4XLP) |

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

# Appendix – validation of kinetic analysis

## Introduction

In this Appendix we describe details of the kinetic analyses validating the results and conclusions presented in the main manuscript. First, we establish that the minimal kinetic scheme required to describe our data is the 3-step sequential kinetic scheme (*Appendix 1—figure 1*). Second, we present the results of confidence contour analysis ('FitSpace' analysis within the context of KinTek Global Kinetic Explorer; *Johnson et al., 2009a*) to validate that the parameter estimation (*Table 1*, main manuscript) is well constrained.

### 1-step

$$R + P \underset{k_{-1}}{\overset{k_1}{\rightleftarrows}} RPo$$

Fluorescence = a[P] + b[RPo] + bkg

### 2-step

$$R + P \underset{k_{-1}}{\overset{k_1}{\rightleftarrows}} RP1 \underset{k_{-2}}{\overset{k_2}{\rightleftarrows}} RPo$$

Fluorescence = a[P] + b[RP1] +c[RPo] + bkg

### 3-step

$$R + P \underset{k_{-1}}{\overset{k_1}{\rightleftarrows}} RP1 \underset{k_{-2}}{\overset{k_2}{\rightleftarrows}} RP2 \underset{k_{-3}}{\overset{k_3}{\rightleftarrows}} RPo$$

Fluorescence = a[P] + b[RP1] +c[RP2] + d[RPo] + bkg

### 4-step

$$R + P \underset{k_{-1}}{\overset{k_1}{\rightleftarrows}} RP1 \underset{k_{-2}}{\overset{k_2}{\rightleftarrows}} RP2 \underset{k_{-3}}{\overset{k_3}{\rightleftarrows}} RP3 \underset{k_{-4}}{\overset{k_4}{\rightleftarrows}} RPo$$

Fluorescence = a[P] + b[RP1] +c[RP2] + d[RP3] + e[RPo] + bkg

**Appendix 1—figure 1.** Kinetic mechanisms. The four sequential kinetic mechanisms considered, along with the equations used to describe the fluorescence signal (the observable).

## Establishing the minimal kinetic scheme

We analyzed the fluorescence progress curves from the Cy3-AP3 promoter (*Figure 3A–C*, *Figure 3—figure supplement 1B–C*) using KinTek Global Kinetic Explorer software, which performs a global fit of a kinetic mechanism to an entire concentration series at once (*Johnson et al., 2009a, 2009b*). In our analysis, we imposed the condition that the same kinetic model applies to each sample (*Eco* holo, *Mbo* holo, *Mbo* holo+RbpA, *Mbo* holo+CarD, *Mbo* holo+RbpA+CarD). In other words, the different RNAPs (*Eco* or *Mbo*) or addition of

factors altered the rate constants associated with the mechanism, but did not alter the basic mechanism. Based on previous analyses (**Buc and McClure, 1985**; **Roe et al., 1985**; **Rutherford et al., 2009**; **Saecker et al., 2011**; **Sclavi et al., 2005**), we considered only sequential (linear) kinetic schemes (1-step, 2-step, 3-step, 4-step; **Appendix 1—figure 1**).

For each sample, the software calculated rate constants, fluorescence scale factors (a, b, c, d, e; **Appendix 1—figure 1**), and $\chi^2$/DOF (DOF = degress of freedom), a 'goodness of fit' parameter. For a perfect fit to the data points, $\chi^2$/DOF = 1. Based on the fitted kinetic constants, we calculated the theoretical RPo $t_{1/2}$ (**Tsodikov and Record, 1999**) for comparison with the experimental values (**Figure 2C**):

$$\frac{1}{k_d} = \frac{1}{k_{-3}} + \frac{1 + K_3}{k_{-2}} + \frac{K_2 + K_2K_3}{k_{-1}} + \frac{1}{k_{-1}} \tag{2}$$

where $t_{1/2} = \ln(2)/k_d$.

For each sample dataset, the 1-step and 2-step models were rejected because: (1) the curves obviously did not fit the data (**Appendix 1—figure 2**); (2) the $\chi^2$/DOF was far from optimal (**Appendix 1—figure 2**; **Appendix 1—tables 1–5**); and (3) in many cases (1-step model: *Mbo* holo+CarD, *Mbo* holo+RbpA+CarD, *Eco* holo; 2-step model: *Mbo* holo+RbpA, *Mbo* holo+CarD, *Mbo* holo+RbpA+CarD), the calculated $t_{1/2}$ of the promoter complexes was inconsistent with the experimental values (**Appendix 1—tables 1–5**). The 4-step model was also rejected, primarily because in every case, one of the fluorescence scale factors ('d', corresponding to the extra intermediate, RP3, introduced in the 4-step pathway) refined to a value near zero (**Appendix 1—tables 1–5**), indicating that adding this intermediate did not contribute to the fitting of the data and often rendered the kinetic constants associated with this intermediate ($k_3$, $k_{-3}$, $k_4$, $k_{-4}$) ill determined.

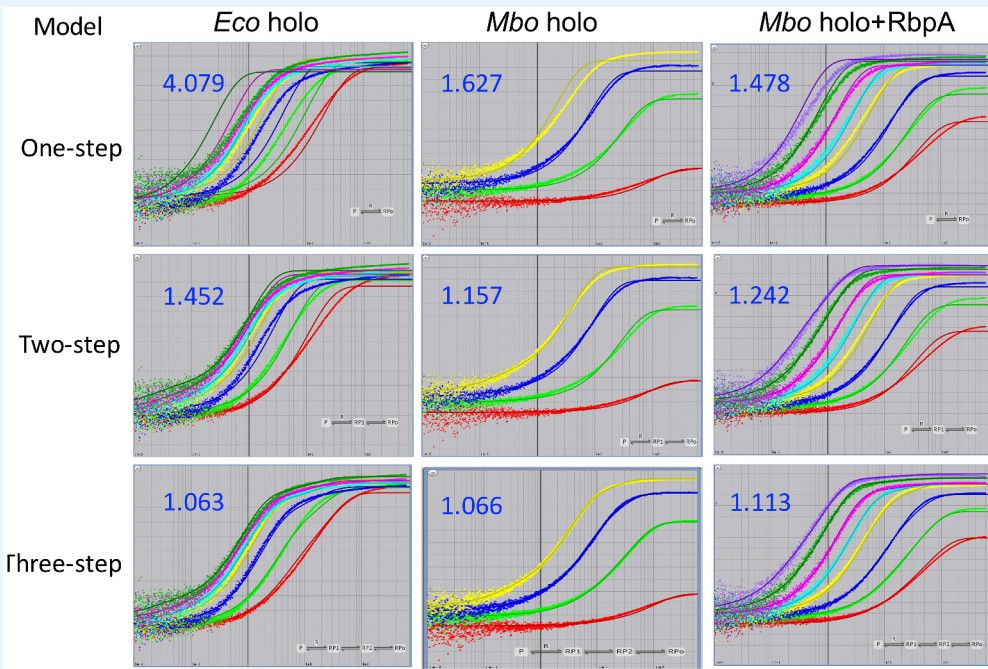

**Appendix 1—figure 2.** Fits of kinetic mechanisms to selected data. Plots showing the fluorescence signal vs. time after mixing *Eco* holo (left column), *Mbo* holo (middle column), or *Mbo* holo+RbpA (right column) with 1 nM Cy3-AP3 promoter (**Figure 3—figure supplement 1A**) in a stopped flow fluorimeter. The experimental data are shown as points. The data were fit using the 1-step (top row), 2-step (middle row), or 3-step (bottom row)

sequential kinetic scheme (*Appendix –figure 1*). The curve fits are shown as solid lines. The blue numbers show the $\chi^2$/DOF for each fit.

**Appendix 1—table 1.** Kinetic model determination for *Mbo* holo on the Cy3-AP3 promoter.

| Model* | 1-step | 2-step | 3-step | 4-step |
|---|---|---|---|---|
| | | *Mbo* holo** | | |
| $k_1$ (M$^{-1}$s$^{-1}$) | $(1.1 \pm 0.4) \times 10^6$ | $(7.8 \pm 1.2) \times 10^6$ | $(9.5 \pm 1.3) \times 10^6$ | $(9.9 \pm 1.7) \times 10^6$ |
| $k_{-1}$ (s$^{-1}$) | $(6.0 \pm 2.2) \times 10^{-3}$ | $2.0 \pm 0.2$ | $2.0 \pm 0.2$ | $1.6 \pm 0.3$ |
| $K_1$ (M$^{-1}$) | $(1.8 \pm 0.9) \times 10^8$ | $(3.9 \pm 0.7) \times 10^6$ | $(4.8 \pm 0.8) \times 10^6$ | $(6.2 \pm 1.6) \times 10^6$ |
| $k_2$ (s$^{-1}$) | | $0.37 \pm 0.02$ | $0.40 \pm 0.02$ | $0.3 \pm 0.07$ |
| $k_{-2}$ (s$^{-1}$) | | $(9.5 \pm 2.3) \times 10^{-3}$ | $0.058 \pm 0.007$ | $0.097 \pm 0.021$ |
| $K_2$ | | $39 \pm 10$ | $6.9 \pm 0.9$ | $3.1 \pm 1.0$ |
| $k_3$ (s$^{-1}$) | | | $0.057 \pm 0.011$ | $0.21 \pm 0.16$ |
| $k_{-3}$ (s$^{-1}$) | | | $0.015 \pm 0.004$ | $2.2 \pm 11.5$ |
| $K_3$ | | | $3.8 \pm 1.3$ | $0.095 \pm 0.504$ |
| $k_4$ (s$^{-1}$) | | | | $3.8 \pm 25.5$ |
| $k_{-4}$ (s$^{-1}$) | | | | $0.048 \pm 0.105$ |
| $K_4$ | | | | $0.025 \pm 0.215$ |
| $\chi^2$/DOF | 1.62744 | 1.15668 | 1.06622 | 1.06737 |
| $k_d$† | $6.0 \times 10^{-3}$ | $9.4 \times 10^{-3}$ | $6.6 \times 10^{-3}$ | $6.0 \times 10^{-3}$ |
| $t_{1/2}$ (min) | 1.9 | 1.2 | 1.7 | 1.9 |
| $t_{1/2}^{exp}$ (min)‡ | ~2 | | | |
| a | 0.48 | 0.55 | 0.30 | 0.65 |
| b | 1.3 | 0.81 | 0.51 | 0.83 |
| c | | 1.4 | 1.2 | 1.8 |
| d | | | 1.2 | 0.026 |
| e | | | | 1.5 |
| bkg | 1.3 | 0.16 | 0.41 | 0.064 |

*Appendix 1—figure 1*.

**The errors listed are the standard errors from the global fit as reported by Kintek Global Kinetic Explorer (Johnson et al., 2009a).

†For the 1-step, 2-step, and 3-step models, the value for $k_d$, the dissociation rate for RPo, was calculated using equation (17) of (*Tsodikov and Record, 1999*):

$$\frac{1}{k_d} = \frac{1}{k_{-3}} + \frac{1 + K_3}{k_{-2}} + \frac{K_2 + K_2 K_3}{k_{-1}} + \frac{1}{k_{-1}} \tag{2}$$

For the 1-step model, $K_2 = K_3 = 0$, $k_{-2} = k_{-3} = \infty$; for the 2-step model, $K_3 = 0$, $k_{-3} = \infty$. For the 4-step model, dissociation was simulated and the result was fit to a single exponential decay to derive $k_d$. The value for $t_{1/2}$ was calculated as $t_{1/2} = \ln(2)/k_d$.

‡The experimental half-life ($t_{1/2}^{exp}$) was determined from promoter lifetime experiments (*Figure 2C*).

**Appendix 1—table 2.** Kinetic model determination for *Mbo* holo+RbpA on the Cy3-AP3 promoter.

| Model* | 1-step | 2-step | 3-step | 4-step |
|---|---|---|---|---|
| | | *Mbo* holo+RbpA** | | |

*Appendix 1—table 2 continued on next page*

*Appendix 1—table 2 continued*

| Model* | 1-step | 2-step | 3-step | 4-step |
|---|---|---|---|---|
| | **Mbo holo+RbpA**[**] | | | |
| $k_1$ ($M^{-1}s^{-1}$) | $(6.4 \pm 0.3) \times 10^6$ | $(1.2 \pm 0.1) \times 10^7$ | $(1.5 \pm 0.1) \times 10^7$ | $(1.6 \pm 0.1) \times 10^7$ |
| $k_{-1}$ ($s^{-1}$) | $(6.5 \pm 1.1) \times 10^{-3}$ | $0.80 \pm 0.07$ | $1.1 \pm 0.1$ | $0.88 \pm 0.09$ |
| $K_1$ ($M^{-1}$) | $(9.9 \pm 1.7) \times 10^8$ | $(1.5 \pm 0.2) \times 10^7$ | $(1.4 \pm 0.2) \times 10^7$ | $(1.8 \pm 0.2) \times 10^7$ |
| $k_2$ ($s^{-1}$) | | $1.1 \pm 0.1$ | $1.4 \pm 0.1$ | $0.81 \pm 0.08$ |
| $k_{-2}$ ($s^{-1}$) | | $0.017 \pm 0.002$ | $0.090 \pm 0.020$ | $0.062 \pm 0.030$ |
| $K_2$ | | $65 \pm 10$ | $16 \pm 4$ | $13 \pm 6$ |
| $k_3$ ($s^{-1}$) | | | $0.046 \pm 0.015$ | $0.59 \pm 0.75$ |
| $k_{-3}$ ($s^{-1}$) | | | $0.011 \pm 0.004$ | $1.6 \pm 0.7$ |
| $K_3$ | | | $4.2 \pm 2.0$ | $0.37 \pm 0.50$ |
| $k_4$ ($s^{-1}$) | | | | $0.091 \pm 0.193$ |
| $k_{-4}$ ($s^{-1}$) | | | | $(9.9 \pm 14) \times 10^{-3}$ |
| $K_4$ | | | | $4.1 \pm 10.2$ |
| $\chi^2$/DOF | 1.47786 | 1.24241 | 1.11268 | 1.10116 |
| $k_d$[†] | $6.4 \times 10^{-3}$ | 0.016 | $6.4 \times 10^{-3}$ | $3.6 \times 10^{-3}$ |
| $t_{1/2}$ (min) | 1.8 | 0.71 | 1.8 | 3.2 |
| $t_{1/2}{}^{exp}$ (min)[‡] | ~1.5 | | | |
| a | 0.099 | 0.038 | 0.16 | 0.65 |
| b | 1.2 | 0.70 | 0.68 | 0.83 |
| c | | 1.1 | 1.3 | 1.8 |
| d | | | 1.3 | 0.026 |
| e | | | | 1.5 |
| bkg | 0.81 | 0.86 | 0.74 | 0.064 |

*****Appendix 1—figure 1**.

**The errors listed are the standard errors from the global fit as reported by Kintek Global Kinetic Explorer (Johnson et al., 2009a).

†TT For the 1-step, 2-step, and 3-step models, the value for $k_d$, the dissociation rate for RPo, was calculated using equation (17) of (*Tsodikov and Record, 1999*):

$$\frac{1}{k_d} = \frac{1}{k_{-3}} + \frac{1 + K_3}{k_{-2}} + \frac{K_2 + K_2 K_3}{k_{-1}} + \frac{1}{k_{-1}} \tag{2}$$

For the 1-step model, $K_2 = K_3 = 0$, $k_{-2} = k_{-3} = \infty$; for the 2-step model, $K_3 = 0$, $k_{-3} = \infty$. For the 4-step model, dissociation was simulated and the result was fit to a single exponential decay to derive $k_d$. The value for $t_{1/2}$ was calculated as $t_{1/2} = \ln(2)/k_d$.

‡The experimental half-life ($t_{1/2}{}^{exp}$) was determined from promoter lifetime experiments (*Figure 2C*).

**Appendix 1—table 3.** Kinetic model determination for *Mbo* holo+CarD on the Cy3-AP3 promoter.

| Model* | 1-step | 2-step | 3-step | 4-step |
|---|---|---|---|---|
| | **Mbo holo+CarD**** | | | |
| $k_1$ ($M^{-1}s^{-1}$) | $(1.3 \pm 0.1) \times 10^7$ | $(5.4 \pm 0.4) \times 10^7$ | $(5.8 \pm 0.5) \times 10^7$ | $(3.8 \pm 0.3) \times 10^7$ |
| $k_{-1}$ ($s^{-1}$) | $(3.2 \pm 1.4) \times 10^{-3}$ | $10 \pm 1$ | $6.2 \pm 1.0$ | $1.4 \pm 0.2$ |

*Appendix 1—table 3 continued on next page*

Appendix 1—table 3 continued

|  | **Mbo holo+CarD**[**] | | | |
|---|---|---|---|---|
| **Model*** | **1-step** | **2-step** | **3-step** | **4-step** |
| $K_1$ (M$^{-1}$) | $(4.1 \pm 1.8) \times 10^9$ | $(5.4 \pm 0.7) \times 10^6$ | $(9.4 \pm 1.7) \times 10^6$ | $(2.7 \pm 0.4) \times 10^7$ |
| $k_2$ (s$^{-1}$) |  | $4.0 \pm 0.3$ | $3.1 \pm 0.2$ | $1.1 \pm 0.2$ |
| $k_{-2}$ (s$^{-1}$) |  | $(2.7 \pm 1.8) \times 10^{-3}$ | $0.067 \pm 0.019$ | $0.049 \pm 0.024$ |
| $K_2$ |  | $1500 \pm 99$ | $46 \pm 14$ | $22 \pm 12$ |
| $k_3$ (s$^{-1}$) |  |  | $0.074 \pm 0.027$ | $0.72 \pm 0.41$ |
| $k_{-3}$ (s$^{-1}$) |  |  | $(2.5 \pm 3.0) \times 10^{-3}$ | $1.4 \pm 0.4$ |
| $K_3$ |  |  | $30 \pm 37$ | $0.51 \pm 0.32$ |
| $k_4$ (s$^{-1}$) |  |  |  | $0.11 \pm 0.06$ |
| $k_{-4}$ (s$^{-1}$) |  |  |  | $(3.9 \pm 5.3) \times 10^{-3}$ |
| $K_4$ |  |  |  | $4.7 \pm 3.9$ |
| $\chi^2$/DOF | 1.34903 | 1.22448 | 1.10877 | 1.1021 |
| $k_d$[†] | $3.2 \times 10^{-3}$ | $2.6 \times 10^{-3}$ | $1.1 \times 10^{-3}$ | $1.2 \times 10^{-3}$ |
| $t_{1/2}$ (min) | 3.6 | 4.3 | 10 | 9.6 |
| $t_{1/2}^{exp}$ (min)[‡] | ~10 |  |  |  |
| a | 0.50 | 0.56 | 0.28 | 0.80 |
| b | 1.5 | 0.75 | 0.45 | 1.1 |
| c |  | 1.5 | 1.2 | 2.6 |
| d |  |  | 1.2 | $5.3 \times 10^{-5}$ |
| e |  |  |  | 1.8 |
| bkg | 0.31 | 0.24 | 0.52 | $2.9 \times 10^{-3}$ |

*Appendix 1—figure 1.

**The errors listed are the standard errors from the global fit as reported by Kintek Global Kinetic Explorer (Johnson et al., 2009a).

†For the 1-step, 2-step, and 3-step models, the value for $k_d$, the dissociation rate for RPo, was calculated using equation (17) of *Tsodikov and Record (1999)*:

$$\frac{1}{k_d} = \frac{1}{k_{-3}} + \frac{1 + K_3}{k_{-2}} + \frac{K_2 + K_2 K_3}{k_{-1}} + \frac{1}{k_{-1}} \tag{2}$$

For the 1-step model, $K_2 = K_3 = 0$, $k_{-2} = k_{-3} = \infty$; for the 2-step model, $K_3 = 0$, $k_{-3} = \infty$. For the 4-step model, dissociation was simulated and the result was fit to a single exponential decay to derive $k_d$. The value for $t_{1/2}$ was calculated as $t_{1/2} = \ln(2)/k_d$.

‡The experimental half-life ($t_{1/2}^{exp}$) was determined from promoter lifetime experiments (*Figure 2C*).

**Appendix 1—table 4.** Kinetic model determination for *Mbo* holo+RbpA+CarD on the Cy3-AP3 promoter.

|  | **Mbo holo+RbpA+CarD**[**] | | | |
|---|---|---|---|---|
| **Model*** | **1-step** | **2-step** | **3-step** | **4-step** |
| $k_1$ (M$^{-1}$s$^{-1}$) | $(2.7 \pm 0.1) \times 10^7$ | $(1.2 \pm 0.2) \times 10^8$ | $(1.1 \pm 0.2) \times 10^8$ | $(5.4 \pm 0.3) \times 10^7$ |
| $k_{-1}$ (s$^{-1}$) | $(6.1 \pm 1.5) \times 10^{-3}$ | $27 \pm 5$ | $20 \pm 5$ | $2.0 \pm 0.4$ |
| $K_1$ (M$^{-1}$) | $(4.4 \pm 1.1) \times 10^9$ | $(4.4 \pm 1.1) \times 10^6$ | $(5.5 \pm 1.7) \times 10^6$ | $(2.7 \pm 0.6) \times 10^7$ |
| $k_2$ (s$^{-1}$) |  | $9.6 \pm 0.4$ | $9.7 \pm 0.5$ | $3.9 \pm 0.5$ |

Appendix 1—table 4 continued on next page

*Appendix 1—table 4 continued*

|  | **Mbo holo+RbpA+CarD[**]** | | | |
|---|---|---|---|---|
| **Model[*]** | **1-step** | **2-step** | **3-step** | **4-step** |
| $k_{-2}$ (s$^{-1}$) |  | $1.1 \times 10^{-7}$ | $0.031 \pm 0.011$ | $0.11 \pm 0.43$ |
| $K_2$ |  | $8.7 \times 10^7$ | $310 \pm 11$ | $35 \pm 140$ |
| $k_3$ (s$^{-1}$) |  |  | $0.13 \pm 0.03$ | $11 \pm 34$ |
| $k_{-3}$ (s$^{-1}$) |  |  | $(1.3 \pm 8.2) \times 10^{-3}$ | $9.0 \pm 30$ |
| $K_3$ |  |  | $100 \pm 63$ | $1.2 \pm 5.6$ |
| $k_4$ (s$^{-1}$) |  |  |  | $0.16 \pm 0.42$ |
| $k_{-4}$ (s$^{-1}$) |  |  |  | $(2.9 \pm 3800) \times 10^6$ |
| $K_4$ |  |  |  | $7.6 \pm 40$ |
| $\chi^2$/DOF | 1.32367 | 1.09473 | 1.05035 | 1.0458 |
| $k_d$[†] | $6.1 \times 10^{-3}$ | $1.1 \times 10^{-7}$ | $2.5 \times 10^{-4}$ | $4.3 \times 10^{-7}$ |
| $t_{1/2}$ (min) | 1.9 | $1.1 \times 10^5$ | 47 | $2.7 \times 10^4$ |
| $t_{1/2}^{exp}$ (min)[‡] | ~30 |  |  |  |
| a | 0.59 | 0.63 | 0.28 | 0.91 |
| b | 1.7 | 0.63 | 0.28 | 0.95 |
| c |  | 1.6 | 1.2 | 4.2 |
| d |  |  | 1.3 | $5.7 \times 10^{-6}$ |
| e |  |  |  | 1.9 |
| bkg | 0.34 | 0.28 | 0.63 | $1.1 \times 10^{-6}$ |

[*]*Appendix 1—figure 1*.

[**]The errors listed are the standard errors from the global fit as reported by Kintek Global Kinetic Explorer (Johnson et al., 2009a).

[†]For the 1-step, 2-step, and 3-step models, the value for $k_d$, the dissociation rate for RPo, was calculated using equation (17) of *Tsodikov and Record (1999)*:

$$\frac{1}{k_d} = \frac{1}{k_{-3}} + \frac{1 + K_3}{k_{-2}} + \frac{K_2 + K_2 K_3}{k_{-1}} + \frac{1}{k_{-1}} \tag{2}$$

For the 1-step model, $K_2 = K_3 = 0$, $k_{-2} = k_{-3} = \infty$; for the 2-step model, $K_3 = 0$, $k_{-3} = \infty$. For the 4-step model, dissociation was simulated and the result was fit to a single exponential decay to derive $k_d$. The value for $t_{1/2}$ was calculated as $t_{1/2} = \ln(2)/k_d$.

c[‡]The experimental half-life ($t_{1/2}^{exp}$) was determined from promoter lifetime experiments (*Figure 2C*).

**Appendix 1—table 5.** Kinetic model determination for *Eco* holo on the Cy3-AP3 promoter.

|  | **Eco holo[**]** | | | |
|---|---|---|---|---|
| **Model[*]** | **1-step** | **2-step** | **3-step** | **4-step** |
| $k_1$ (M$^{-1}$s$^{-1}$) | $(9.9 \pm 0.7) \times 10^6$ | $(1.2 \pm 0.1) \times 10^8$ | $(1.2 \pm 0.1) \times 10^8$ | $(8.5 \pm 0.6) \times 10^7$ |
| $k_{-1}$ (s$^{-1}$) | $(2.7 \pm 2.2) \times 10^{-3}$ | $4.6 \pm 0.7$ | $3.4 \pm 0.5$ | $1.1 \pm 0.1$ |
| $K_1$ (M$^{-1}$) | $(3.7 \pm 3.0) \times 10^9$ | $(2.6 \pm 0.5) \times 10^7$ | $(3.5 \pm 0.6) \times 10^7$ | $(7.7 \pm 0.9) \times 10^7$ |
| $k_2$ (s$^{-1}$) |  | $1.0 \pm 0.01$ | $1.2 \pm 0.03$ | $0.39 \pm 0.04$ |
| $k_{-2}$ (s$^{-1}$) |  | ~0 | $0.11 \pm 0.01$ | $0.024 \pm 0.023$ |
| $K_2$ |  | - | $11 \pm 1$ | $16 \pm 16$ |
| $k_3$ (s$^{-1}$) |  |  | $0.083 \pm 0.015$ | $0.66 \pm 0.25$ |

*Appendix 1—table 5 continued on next page*

*Appendix 1—table 5 continued*

| Model* | Eco holo** 1-step | 2-step | 3-step | 4-step |
|--------|--------|--------|--------|--------|
| $k_{-3}$ (s$^{-1}$) | | | ~0 | $0.65 \pm 0.19$ |
| $K_3$ | | | - | $1.0 \pm 0.5$ |
| $k_4$ (s$^{-1}$) | | | | $0.048 \pm 0.119$ |
| $k_{-4}$ (s$^{-1}$) | | | | ~0 |
| $K_4$ | | | | - |
| $\chi^2$/DOF | 4.07941 | 1.45273 | 1.0631 | 1.04588 |
| $k_d$† | $2.7 \times 10^{-3}$ | ~0 | ~0 | ~0 |
| $t_{1/2}$ (min) | 4.3 | - | - | - |
| $t_{1/2}{}^{exp}$ (min)‡ | >>60 | | | |
| a | 0.53 | 0.55 | 0.52 | 0.60 |
| b | 1.5 | 0.81 | 0.62 | 0.72 |
| c | | 1.4 | 1.4 | 2.9 |
| d | | | 1.4 | $6.1 \times 10^{-4}$ |
| e | | | | 1.5 |
| bkg | 0.32 | 0.16 | 0.28 | 0.20 |

*__Appendix 1—figure 1__.

**The errors listed are the standard errors from the global fit as reported by Kintek Global Kinetic Explorer (Johnson et al., 2009a).

†For the 1-step, 2-step, and 3-step models, the value for $k_d$, the dissociation rate for RPo, was calculated using equation (17) of __Tsodikov and Record (1999)__:

$$\frac{1}{k_d} = \frac{1}{k_{-3}} + \frac{1 + K_3}{k_{-2}} + \frac{K_2 + K_2 K_3}{k_{-1}} + \frac{1}{k_{-1}} \qquad (2)$$

For the 1-step model, $K_2 = K_3 = 0$, $k_{-2} = k_{-3} = \infty$; for the 2-step model, $K_3 = 0$, $k_{-3} = \infty$. For the 4-step model, dissociation was simulated and the result was fit to a single exponential decay to derive $k_d$. The value for $t_{1/2}$ was calculated as $t_{1/2} = \ln(2)/k_d$.

‡The experimental half-life ($t_{1/2}{}^{exp}$) was determined from promoter lifetime experiments (__Figure 2C__).

The 3-step sequential model applied to each data set well: it visually fit the data (__Figure 3A–C__, __Figure 3—figure supplement 1B–C__), it gave $\chi^2$/DOF values close to 1 (__Appendix 1—figure 2__; __Appendix 1—tables 1–5__), and for every sample, the calculated $t_{1/2}$ matched well with the experimental values (__Appendix 1—tables 1–5__). Thus, for each sample, a concentration series was collected on a given day and fitted to the 3-step model. Concentration series collected on separate days were fitted separately, yielding similar constants. The constants derived in this way were then averaged, giving rise to the average values and standard errors listed in __Supplementary file 4__.

**Validation of kinetic parameter estimation** (see __Johnson et al., 2009a__ for a more detailed discussion)

Like other programs for fitting data by nonlinear regression, KinTek Global Kinetic Explorer provides a covariance matrix from which standard errors are calculated to give estimates of the confidence intervals on fitted parameters (__Johnson et al., 2009a__). However, standard errors computed in this way underestimate the true error, and may not be valid at all. The fitting process assumes that all experimental errors are uncorrelated and normally

distributed, which is not necessarily the case for complex kinetic models, where sets of parameters can be correlated in ways that are not obvious. Moreover, the covariance matrix is not valid when the parameters are underconstrained by the data. The practical manifestation of this is that the set of parameter values determined to be the best fit (i.e. a minimum in the sum square error, or SSE, with the experimental data) may not be unique; sets of parameters that are correlated may vary systematically over a wide range and give rise to an equally good fit to the data given appropriate adjustments of the remaining parameters. For this reason, it is difficult to evaluate whether the set of parameters are adequately constrained by the data.

In contour confidence analysis ('FitSpace' calculations in the KinTek Global Kinetic Explorer software; (*Johnson et al., 2009a*), a given pair of parameters (x, y; the test parameters) are varied systematically around the area of their best fit. At each pair of x, y values of the test parameters, the experimental data are fit by allowing all of the other parameters to vary while holding the test parameters fixed. After convergence to the best fit, the sum of squared errors (SSE) is computed. In this way, a profile of the SSE over the grid of tested x, y values is constructed. This process is then repeated for all possible pairs of parameters. This defines the extent to which pairs of parameters are correlated and individual parameters are constrained by the data without making assumptions regarding the values of the other parameters.

The FitSpace calculations for some of the fits to individual concentration series datasets (used to derive the values listed in *Supplementary file 4*) revealed that some of the parameters for some of the fits were not well constrained. We imposed additional constraints on the data to reduce the degress of freedom for the fitted parameters:

1. First, we combined the independent trials for each sample and analyzed them together rather than separately (all samples).
2. Second, for samples where dissociation data were available (*Mbo* holo and *Mbo* holo +RbpA; *Figure 3—figure supplement 1D–E*), we included those data in the combined analysis.
3. Third, for *Mbo* holo, we included the results of the 2-AP experiment (*Figure 3D*) in the combined analysis.
4. Fourth, for *Eco* holo, the unconstrained parameter was $k_{-3}$, which was very small in magnitude relative to the other rate constants and could vary over several orders of magnitude without affecting the fit. In this case, we fixed the value of $k_{-3}$ to zero.
5. Finally, for the other samples where dissociation data were not available (*Mbo* holo+CarD, *Mbo* holo+RbpA+CarD), we used an iterative procedure to require that the fitted kinetic parameters yielded the observed experimental $t_{1/2}$ for each sample and fixed $k_{-3}$, as detailed here:
   We determined that the calculated RPo $t_{1/2}$ (using *Equations 2 and 3*, main manuscript) was most sensitive to the value of $k_{-3}$ (compared to all the other parameters). Therefore, for each dataset (*Mbo* holo+CarD, *Mbo* holo+RbpA+CarD), we adjusted the value of $k_{-3}$ until the calculated RPo $t_{1/2}$ was equal to the experimentally determined value (to within round-off error). We then fixed $k_{-3}$ and refined the other constants against the fluorescence data. We iteratively adjusted $k_{-3}$ and refined the other values so that the final calculated RPo $t_{1/2}$ was equal to the experimentally determined value, and the refinement of the other constants stabilized (i.e. the values didn't change during refinement). Note that since the initial (unrestrained) refinements resulted in calculated RPo $t_{1/2}$ values that were already close to the experimental values (*Supplementary file 4*), this procedure did not require dramatic changes to the constants but resulted in a final refinement in which $k_{-3}$ was fixed.

The final refinement and FitSpace calculations for each sample are described below:

## *Eco* holo

For *Eco* holo, we only collected one concentration series (*Supplementary file 3*; *Figure 3A*). FitSpace calculations on the unconstrained fit revealed that the parameter $k_{-3}$ was poorly constrained by the data; the value of $k_{-3}$ could vary from zero over several orders of

magnitude with little effect on the fit of the remaining parameters. The experimental determination of the RPo $t_{1/2}$ for *Eco* holo on AP3 could only provide a lower bound, since over the time course of up to 2 hr experiments, essentially no dissociation was observed (**Davis et al., 2015**). We listed the $t_{1/2}$ as >>60 min (**Supplementary file 4**). We therefore fixed $k_{-3}$ to zero. The $\chi^2$/DOF value was unaltered (1.063 whether $k_{-3}$ was fixed at zero or not; **Appendix 1—figure 2**). The resulting FitSpace calculation showed that the remaining parameters ($k_1$, $k_{-1}$, $k_2$, $k_{-2}$, $k_3$) were constrained (**Appendix 1—figure 3**).

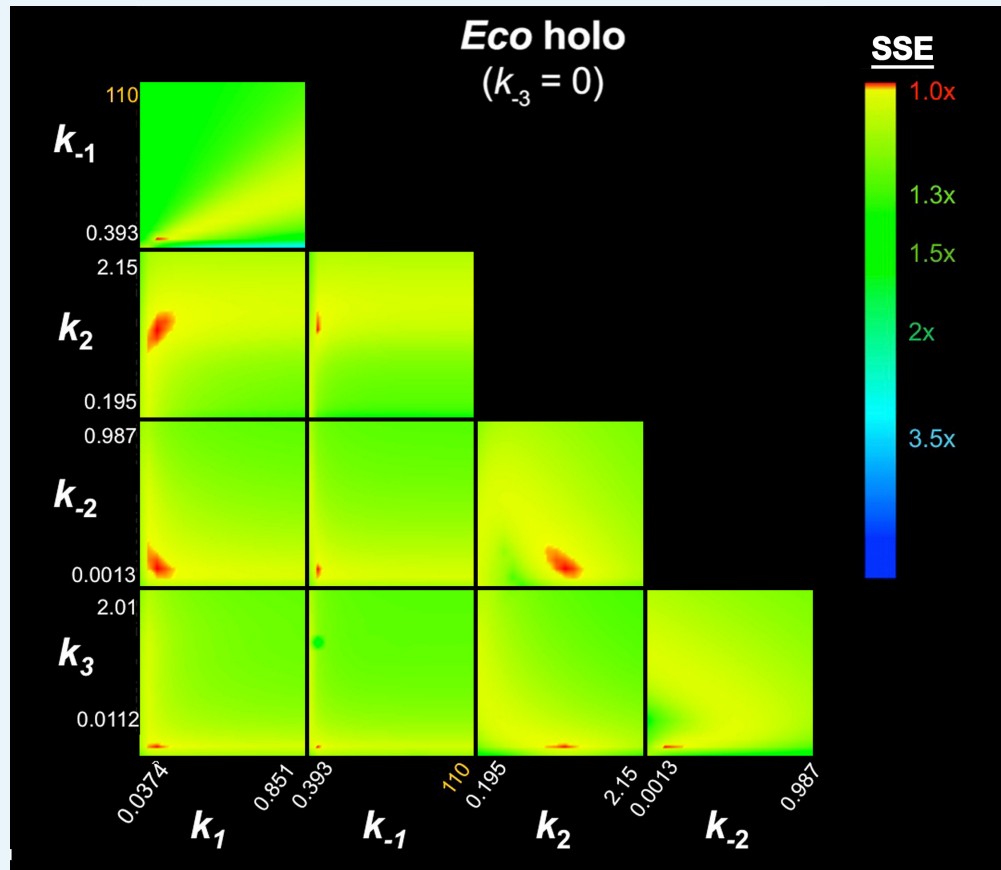

**Appendix 1—figure 3.** FitSpace calculation for *Eco* holo ($k_{-3} = 0$). The results of the contour confidence analysis for *Eco* holo (parameters listed in **Table 1**) with $k_{-3}$ fixed at 0. Each panel represents a grid where two kinetic parameters were varied systematically. For example, the upper left panel shows the result of systematically varying $k_1$ from $3.74 \times 10^7$ to $8.51 \times 10^8$ M$^{-1}$s$^{-1}$ (x-axis) and $k_{-1}$ from 0.393 to 110 s$^{-1}$ (y-axis). At each point, the values of $k_1$ and $k_{-1}$ were fixed, then the other parameters ($k_2$, $k_{-2}$, $k_3$) were refined against the data (the fluorescence progress curves; **Figure 3A**). The resulting sum squared error (SSE) is normalized by the minimum SSE (SSE$_{min}$) and plotted as a heat map. The minimum SSE (normalized by itself, so with a value of 1) is red in the heat map, higher SSE values are color-coded as shown. Each panel shows that the optimal pair of parameter values are restricted to a small red area, indicating the parmeters are constrained by the data.

### *Mbo* holo

The datasets combined and analyzed globally included three independent association concentration series (such as shown in **Figure 3B**, **Appendix 1—figure 2**), one dissociation experiment (**Figure 3—figure supplement 1D**), and the 2-AP experiment (**Figure 3D**). The global refinement weight for each of the association datasets was set to 1, while the

weight for the dissociation and 2-AP experiment was set to 2. The FitSpace calculation showed that the fitted kinetic parameters were constrained (*Appendix 1—figure 4*).

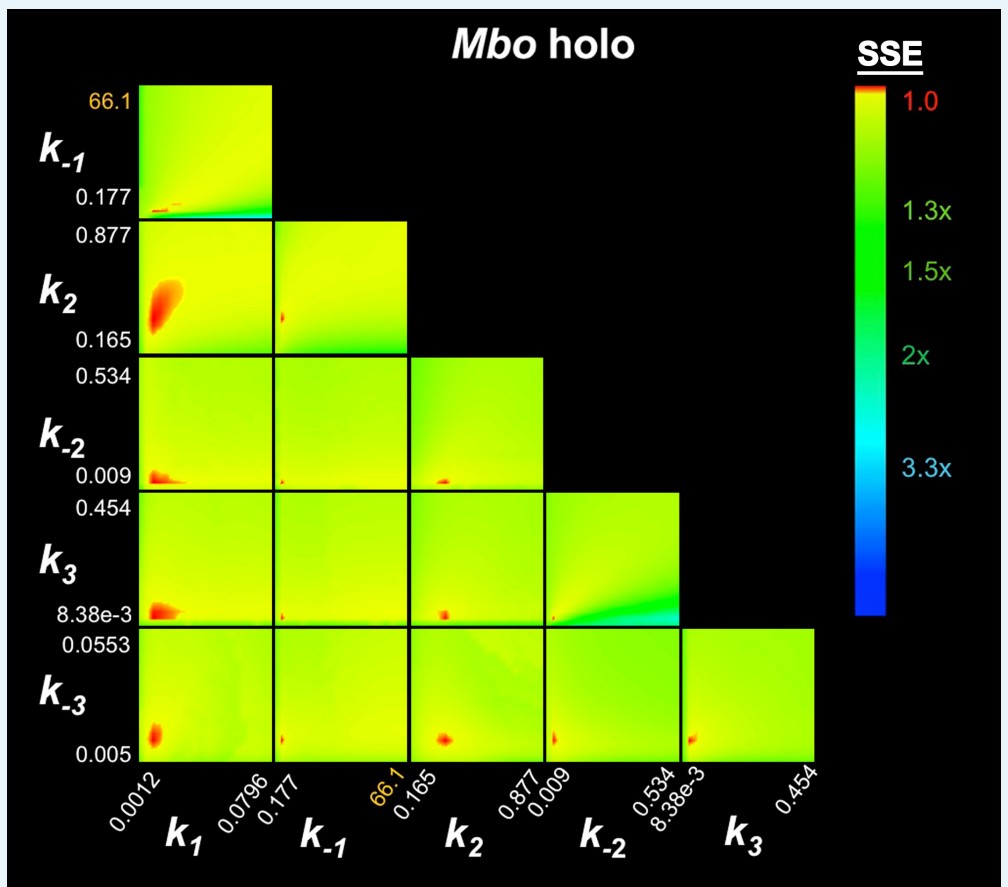

**Appendix 1—figure 4.** FitSpace calculation for *Mbo* holo.

## *Mbo* holo+RbpA

The datasets combined and analyzed globally included two independent association concentration series (such as shown in *Figure 3—figure supplement 1B*, *Appendix 1—figure 2*) and one dissociation experiment (*Figure 3—figure supplement 1E*). The global refinement weight for each of the association datasets was set to 1, while the weight for the dissociation experiment was set to 2. The FitSpace calculation showed that the fitted kinetic parameters were constrained (*Appendix 1—figure 5*).

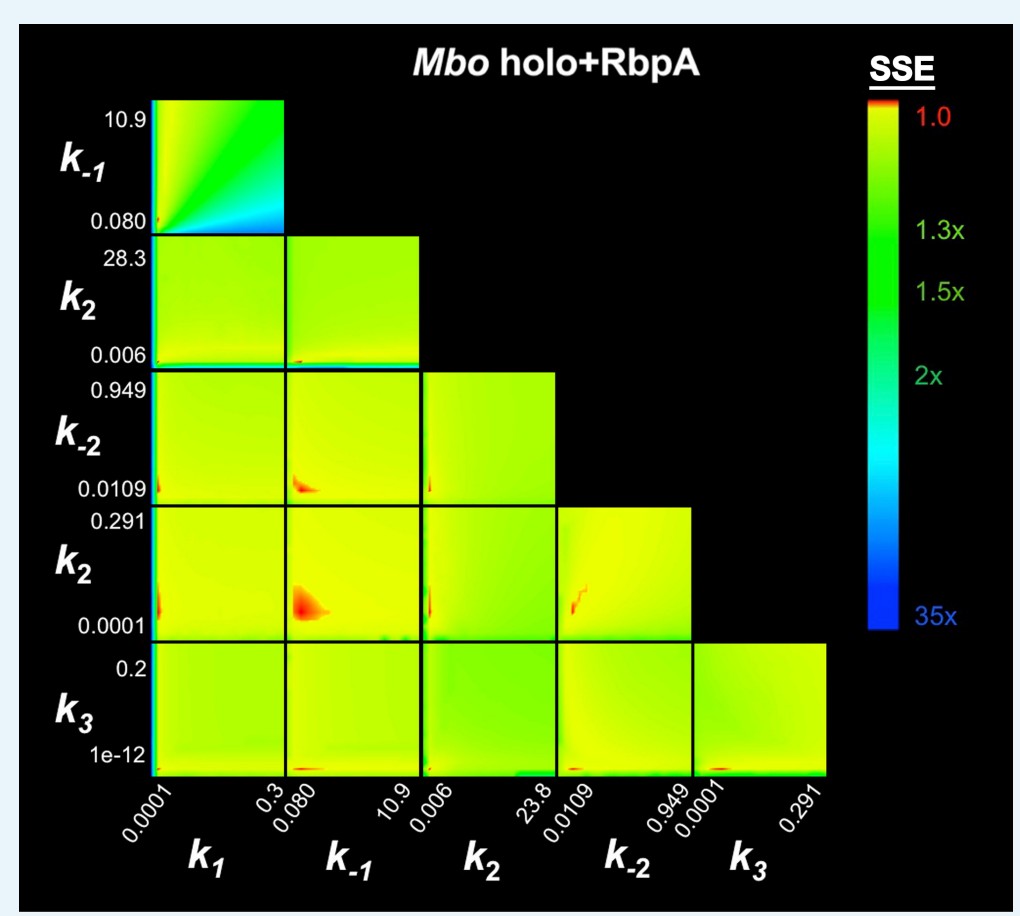

**Appendix 1—figure 5.** FitSpace calculation for *Mbo* holo+RbpA.

### *Mbo* holo+CarD

The datasets combined and analyzed globally included two independent association concentration series (such as shown in *Figure 3—figure supplement 1C*). An iterative procedure was followed wherein $k_{-3}$ was adjusted so that the calculated RPo $t_{1/2}$ matched the experimental value of 10 min (*Figure 2C*). The value of $k_{-3}$ was then fixed, and the remaining parameters were refined. This procedure was repeated until the refinement stabilized on a set of values yielding a calculated RPo $t_{1/2}$ as close to 10 min as possible. This led to the fixed value of $k_{-3}=2.3\times10^{-3}$ s$^{-1}$. The FitSpace calculation showed that the remaining kinetic parameters were constrained (*Appendix 1—figure 6*).

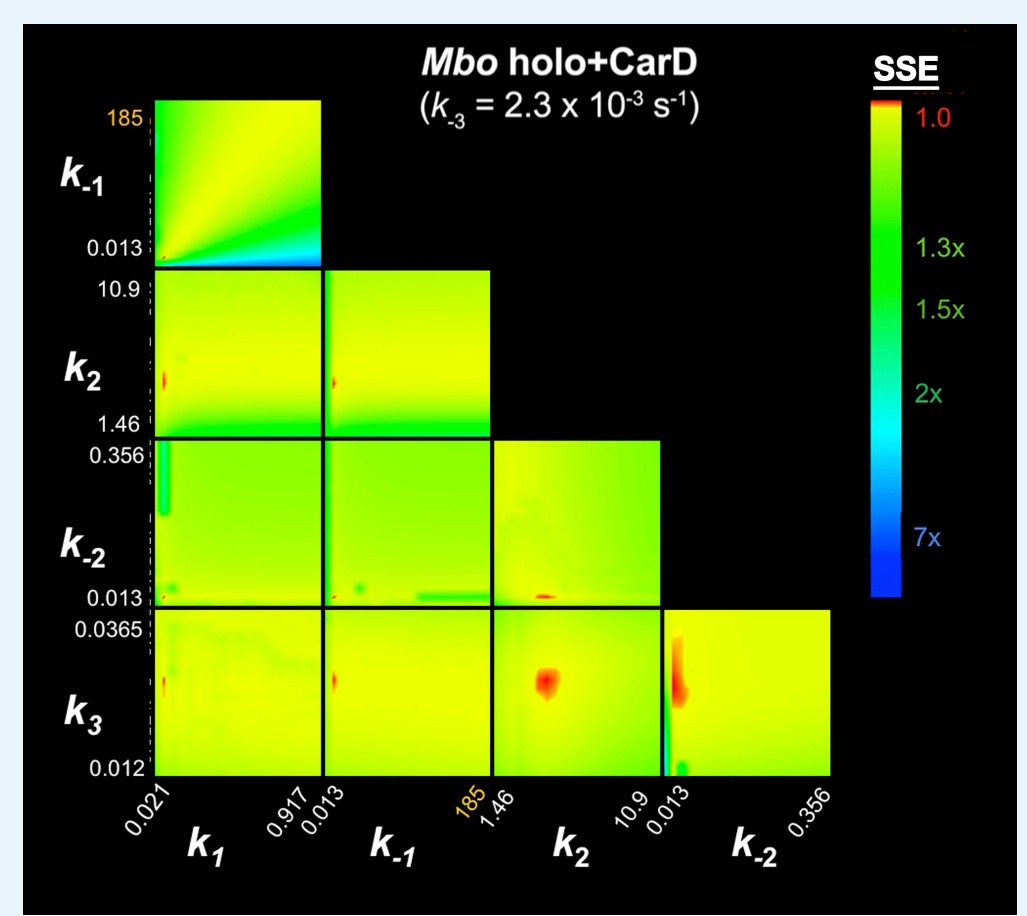

**Appendix 1—figure 6.** FitSpace calculation for *Mbo* holo+CarD ($k_{-3} = 2.3 \times 10^{-3}$ s$^{-1}$).

### *Mbo* holo+RbpA+CarD

The datasets combined and analyzed globally included two independent association concentration series (such as shown in **Figure 3C**). An iterative procedure was followed wherein $k_{-3}$ was adjusted so that the calculated RPo $t_{1/2}$ matched the experimental value of 30 min (**Figure 2C**). The value of $k_{-3}$ was then fixed, and the remaining parameters were refined. This procedure was repeated until the refinement stabilized on a set of values yielding a calculated RPo $t_{1/2}$ as close to 30 min as possible. This led to the fixed value of $k_{-3}=1.3 \times 10^{-3}$ s$^{-1}$. The FitSpace calculation showed that the remaining fitted kinetic parameters were constrained with the exception of $k_1$ (**Appendix 1—figure 7**).

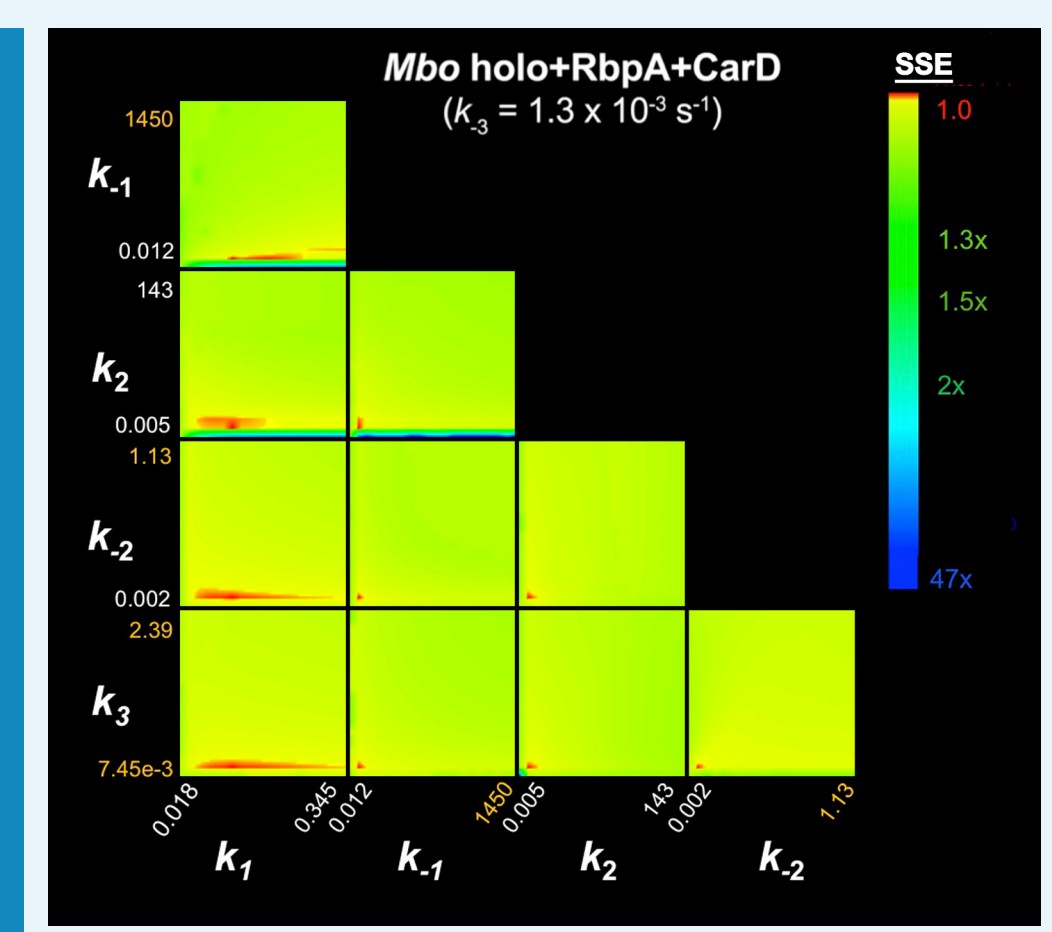

**Appendix 1—figure 7.** FitSpace calculation for *Mbo* holo+RbpA+CarD ($k_{-3} = 1.3 \times 10^{-3}$ s$^{-1}$).

