## [Decision Letter]

Thank you for submitting your article "Structure and function of the mycobacterial transcription initiation complex with the essential regulator RbpA" for consideration by *eLife*. Your article has been reviewed by two peer reviewers, including Ann Hochschild (Reviewer #2), and the evaluation has been overseen by a Reviewing Editor and Richard Losick as the Senior Editor.

The reviewers have discussed the reviews with one another and the Reviewing Editor has drafted this decision to help you prepare a revised submission.

Summary:

This is a thorough, definitive paper that addresses a number of questions about Mycobacterial promoters and their regulation by the transcription factors RbpA and CarD. It reports a high resolution crystal structure (2.76Å) of RpbA in complex with mycobacterial RNA polymerase, a detailed kinetic analysis deriving from extensive experiments using a fluorescence assay that reports on open complex formation, structure-function analysis of RbpA and RpbA mutants on transcription, a model for how RpbA and CarD work together, RNA-seq gene expression analysis addressing the impact of RpbA and its domains genome-wide, and microscopic analysis of cell morphology expressing the various RpbA mutants. The results explain how two transcription factors, RpbA and CarD, both can act on the transcription complex simultaneously and can synergize. The findings are for the most part clear and compelling, though the in vivo results with the RbpA R79A mutant are somewhat puzzling and may warrant further discussion (see below). The manuscript might also benefit from some tightening/shortening.

Essential revisions:

1) Subsection “Full-length RbpA is essential for normal growth of *Msm*”: The viability of the RbpA-R79A mutant is somewhat surprising, especially given the repressive effects of this mutant as seen in vitro (Figure 2). Do the authors believe that the in vitro assays are more sensitive and that, in vivo, the BL can still engage the DNA phosphate backbone even in the absence of the R79 side chain? Some discussion of this apparent discrepancy would be helpful (see also comment 4, below).

2) Subsection “Full-length RbpA is essential for normal growth of *Msm*” and Figure 5: The cell morphology images are difficult to interpret; the differences the authors highlight in the text are hard to appreciate visually. For just one example, the Hoechst stain in the upper panel of the RbpA^CD-BL-SID^ sample reveals two foci that look very similar to the apparently aberrant structures in the top panel of the RbpA^BL-SID^ sample (designated condensed nucleoid). This panel should be improved or eliminated.

3) Subsection “Differential effects of RbpA domains on gene expression in vivo”: How do the authors interpret this observation – that the loss of the NTT results in changes in gene expression that are not seen when both the NTT and the CD are removed? It is as if removal of the CD suppresses the effects of removing the NTT.

4) Subsection “RP2”, end of third paragraph: The suggestion that the interaction of RbpA residue R79 with the nt strand -13 phosphate fulfills an essential anchoring role, while mechanistically satisfying, must still be reconciled with the viability of the RbpA-R79A allele, especially given that most *Mtb* promoters lack a -35 element.

5) Abstract. "Unlike *E. coli* RPo, Mycobacterial RPo is highly reversible." Later in the paper the authors go on to show that the two RNAPs differ in the stability of the complexes they form even on the same promoter, but at this point the blanket statement is misleading. *E. coli* also forms highly reversible complexes on certain promoters.

6) Subsection “The RbpA-R79/DNA interaction is critical for RbpA in vitro function”, first paragraph, and first sentence in Discussion. Earlier ChIP-seq studies indicated that CarD is present on most/all σ A-dependent promoters, and the work presented here shows that RbpA works in conjunction with RbpA on some promoters. However, the sentence implies that RbpA also functions genome-wide. Is that what the authors mean to say? If not, please rephrase, and if so, please describe the data that justify this statement.

7) Subsection “The RbpA-R79/DNA interaction is critical for RbpA in vitro function”, third paragraph. Figure 2. I found this figure difficult to understand, and specifically I could not figure out where the statement "decreased more than 10-fold" came from. What is the Y-axis?

8) Subsection “RbpA and CarD Affect Distinct Kinetic Steps of RPo Formation on AP3”, last sentence in first paragraph. Although it is true that there is a real advantage of studying kinetics without competitors like heparin, and they have included the word "mostly", there have been many studies of association kinetics without competitors. Maybe it would be better to just clearly state why it is best to study kinetics without perturbants, without implying that such studies are rare.

9) The kinetics section in the Results seems a bit long-winded. Can it be condensed for readability? At the same time, it would be helpful for the casual reader to state that the energy profile (i.e. Figure 3) is simply a replot using the kinetic constants derived in parts A-E, using software described in the Johnson et al. papers. The extensive "Appendix" is overwhelming. [Supplementary-material SD4-data] does not fit on the page in the PDF.

10) Subsection “RP1”, second paragraph: discrepancy with Rammohan et al. 2015. Are there other explanations for the discrepancy besides "buffer composition"? I understand that glutamate buffers would tend to stabilize complexes and alter the overall rates across the board, but would this really change the conclusions about the basic mechanism?

11) Subsection “RbpA function in vivo” and last line in Figure 5 legend ("See supplemental dataset for gene lists corresponding to each color class"). I assume the authors are referring to the Excel files provided. It is nice that the authors are providing these data for the community. Given the massive amounts of data in this paper, I'm not asking for more, but there must be something interesting about specific transcripts that would be useful to point out.

---

## [Author Response]

*Essential revisions:*

*1) Subsection “Full-length RbpA is essential for normal growth of Msm”: The viability of the RbpA-R79A mutant is somewhat surprising, especially given the repressive effects of this mutant as seen in vitro (Figure 2). Do the authors believe that the in vitro assays are more sensitive and that, in vivo, the BL can still engage the DNA phosphate backbone even in the absence of the R79 side chain? Some discussion of this apparent discrepancy would be helpful (see also comment 4, below).*

We have added the following to the Discussion section:

“In our in vitroanalyses, RbpA^R79A^ by itself appeared to have lost all RbpA function in activating transcription and counteracted the activity of CarD when both factors were present (Figure 2, [Supplementary-material SD7-data]). […] It seems likely that other promoters are regulated by RbpA^R79A^ in vivo, possibly through the action of additional conserved basic residues on the BL such as K76 (Figure 1—figure supplement 1).”

*2) Subsection “Full-length RbpA is essential for normal growth of Msm” and Figure 5: The cell morphology images are difficult to interpret; the differences the authors highlight in the text are hard to appreciate visually. For just one example, the Hoechst stain in the upper panel of the RbpA^CD-BL-SID^ sample reveals two foci that look very similar to the apparently aberrant structures in the top panel of the RbpA^BL-SID^ sample (designated condensed nucleoid). This panel should be improved or eliminated.*

We have quantitated the nuclear morphology:

“We scored nucleoid morphology as either condensed or diffuse in WT (n=208) and RbpA^BL-SID^ (n=248) and observed that condensed nucleoids were present in 2.4% of WT cells, whereas 25% of RbpA^BL-SID^ nucleoids were condensed (Figure 5).”

We have also included a histogram of the results in a revised Figure 5 and have modified the Figure 5 legend accordingly.

*3) Subsection “Differential effects of RbpA domains on gene expression in vivo”: How do the authors interpret this observation – that the loss of the NTT results in changes in gene expression that are not seen when both the NTT and the CD are removed? It is as if removal of the CD suppresses the effects of removing the NTT.*

We think this is a misunderstanding due to some confusing wording. We have clarified:

Previously: “First, despite the complete complementation by RbpA^CD-BL-SID^ for viability, growth, and cell morphology, loss of the NTT had clear effects on gene expression that were not found in RbpA^BL-SID^, including both overexpressed and underexpressed genes (see red and blue dots, Figure 5).”

Revised: “First, despite the complete complementation by RbpA^CD-BL-SID^ for viability, growth, and cell morphology, loss of the NTT had clear effects on gene expression that were distinct from RbpA^BL-SID^, including both overexpressed and underexpressed genes (see red and blue dots, Figure 5).”

*4) Subsection “RP2”, end of third paragraph: The suggestion that the interaction of RbpA residue R79 with the nt strand -13 phosphate fulfills an essential anchoring role, while mechanistically satisfying, must still be reconciled with the viability of the RbpA-R79A allele, especially given that most Mtb promoters lack a -35 element.*

See response to point 1.

*5) Abstract. "Unlike E. coli RPo, Mycobacterial RPo is highly reversible." Later in the paper the authors go on to show that the two RNAPs differ in the stability of the complexes they form even on the same promoter, but at this point the blanket statement is misleading. E. coli also forms highly reversible complexes on certain promoters.*

We have rewritten the Abstract and this statement is removed.

*6) Subsection “The RbpA-R79/DNA interaction is critical for RbpA* in vitro *function”, first paragraph, and first sentence in Discussion. Earlier ChIP-seq studies indicated that CarD is present on most/all σ A-dependent promoters, and the work presented here shows that RbpA works in conjunction with RbpA on some promoters. However, the sentence implies that RbpA also functions genome-wide. Is that what the authors mean to say? If not, please rephrase, and if so, please describe the data that justify this statement.*

We have added a paragraph to the Introduction explaining why we think CarD and RbpA function together:

“ChIP-seq studies indicate CarD is present at most, if not all s^A^ promoters in mycobacteria (Srivastava et al., 2013). […] Thus, the structural mechanism for CarD and RbpA function must be compatible with the two transcription activators acting simultaneously on the mycobacterial TIC.”

*7) Subsection “The RbpA-R79/DNA interaction is critical for RbpA in vitro function”, third paragraph. Figure 2. I found this figure difficult to understand, and specifically I could not figure out where the statement "decreased more than 10-fold" came from. What is the Y-axis?*

The y-axis title for Figure 2, which was inadvertantly left out, has been added, and additional explanation of the normalization procedure has been added to the Figure legend. The transcription activity for each promoter was normalized with respect to holo activity on that promoter (this is because transcription on the AP3anti-35 promoter was so weak compared to the other promoters, the plot of absolute values is difficult to see what is going on with AP3anti-35. Because of the normalization procedure, this figure does not illustrate the transcription activity of each promoter compared to the other promoters. The abortive transcription gels in Figure 2—figure supplement 1 also do not illustrate this because the assays for the different promoters were conducted with different specific activities of radioactivity. We have added Figure 2—figure supplement 1, which shows transcription on AP3 and AP3anti-35 under conditions where they can be directly compared – this figure shows how holo on AP3anti-35 is ~10-fold weaker than holo on AP3.

*8) Subsection “RbpA and CarD Affect Distinct Kinetic Steps of RPo Formation on AP3”, last sentence in first paragraph. Although it is true that there is a real advantage of studying kinetics without competitors like heparin, and they have included the word "mostly", there have been many studies of association kinetics without competitors. Maybe it would be better to just clearly state why it is best to study kinetics without perturbants, without implying that such studies are rare.*

This is a misunderstanding – the word ‘perturbative’ here was meant to refer to non-real time approaches for measuring RPo formation kinetics (where the reaction had to be perturbed in order to measure something). We have simply deleted the word ‘perturbative’ (to avoid confusion), so the sentence now reads:

“This pathway (Equation 1) has been studied using mostly approaches that require the RPo reactions to be halted, then probed […]”

*9) The kinetics section in the Results seems a bit long-winded. Can it be condensed for readability? At the same time, it would be helpful for the casual reader to state that the energy profile (i.e. Figure 3) is simply a replot using the kinetic constants derived in parts A-E, using software described in the Johnson et al. papers. The extensive "Appendix" is overwhelming. [Supplementary-material SD4-data] does not fit on the page in the PDF.*

We have edited the kinetics section to make it easier to digest by: 1) shortening it a bit, and 2) reorganizing it and breaking it up with section headings.

Appendix – It is important to point out that this type of analysis, where all of the forward and reverse rate constants for the entire kinetic mechanism of RPo formation have been determined, is done here for the first time. It is critical that others understand exactly what was done here. Moreover, in this type of fitting exercise the potential for ‘overfitting’, i.e. assigning values to parameters that are not constrained or not unique, is high. It is essential that the methods and the results be rigorously validated. Therefore, it is essential that the material presented in the Appendix be included with the manuscript. We realize not everyone will be interested in the material presented in the Appendix and may be overwhelmed, but that is precisely why it is included as an Appendix. Papers such as this that present complex analyses often contain Appendices to explain the details of the analyses, and when they do not it is often very difficult or impossible to understand how the analysis was actually done.

[Supplementary-material SD4-data] has been reformatted and the main table fits on one page.

*10) Subsection “RP1”, second paragraph: discrepancy with Rammohan et al. 2015. Are there other explanations for the discrepancy besides "buffer composition"? I understand that glutamate buffers would tend to stabilize complexes and alter the overall rates across the board, but would this really change the conclusions about the basic mechanism?*

The reviewer points out a valid concern: overall we do not know enough about the steps and the detailed conformational changes involved to make a convincing case that buffer composition resulted in a difference between our results and theirs. Detailed examination of the previously published work (Rahhoman et al. 2015) reveals that the authors' proposal that CarD destabilizes RPc (relative to RPc without CarD) is not supported by their analysis. Comparing these relative stabilities requires the binding constant for forming RPc from R*CarD + P: this equilibrium was not part of their model. We believe the conclusion about the destabilization of RPc by CarD arose from the interpretation of their data in the framework of a two-step kinetic model:

R + P = RPc = RPo

while we showed rigorously here that a three-step kinetic scheme is necessary to account for the data (Appendix).

We have added a paragraph (–subsection “RP1”, last paragraph) to outline how our approach differs from the previously published one and its key advantages in regards to the following: analyzing the data without having to interpret multi- exponentials, working at saturating concentrations of the regulators, interpreting the results in the context of the 3-step kinetic model (shown here to apply) instead of the 2-step model, and comparing resulting parameters of the fit (kd calculated from microscopic rates) to independently, experimentally determined values of kd.

*11) Subsection “RbpA function in vivo” and last line in Figure 5 legend ("See supplemental dataset for gene lists corresponding to each color class"). I assume the authors are referring to the Excel files provided. It is nice that the authors are providing these data for the community. Given the massive amounts of data in this paper, I'm not asking for more, but there must be something interesting about specific transcripts that would be useful to point out.*

We attempted to segregate the different classes of RbpA-regulated transcripts into functional classes, but no meaningful patterns emerged. Further analysis along these lines is ongoing.